

# DiuSST: A conceptual model of diurnal warm layers for idealized atmospheric simulations with interactive SST

Reyk Börner[1,2], Jan O. Haerter[1,3,4,5], and Romain Fiévet[1,6]

[1]Niels Bohr Institute, University of Copenhagen, Copenhagen, Denmark
[2]Department of Mathematics and Statistics, University of Reading, Reading, UK
[3]Integrated Modeling, Leibniz Center for Tropical Marine Research, Bremen, Germany
[4]Physics and Earth Sciences, Constructor University Bremen, Bremen, Germany
[5]Department of Physics and Astronomy, University of Potsdam, Potsdam, Germany
[6]Scientific Computing Lab, Max Planck Institute for Meteorology, Hamburg, Germany

**Correspondence:** Reyk Börner (reyk.boerner@reading.ac.uk)

**Abstract.** The diurnal variability of sea surface temperature (SST) may play an important role for cloud organization above the tropical ocean, with implications for precipitation extremes, storminess, and climate sensitivity. Recent cloud-resolving simulations demonstrate how imposed diurnal SST oscillations can strongly, and delicately, impact mesoscale convective organization. In spite of this nuanced interaction, many idealized modeling studies of tropical convection either assume a constant, homogeneous SST or, in case of a responsive sea surface, represent the upper ocean by a slab with fixed thickness. Here we show that slab ocean models with constant heat capacity fail to capture the wind-dependence of observed diurnal sea surface warming. To alleviate this shortcoming, we present a simple, yet explicitly depth-resolved model of upper-ocean temperature dynamics under atmospheric forcing. Our modular scheme describes turbulent mixing as diffusion with a wind-dependent diffusivity, in addition to a bulk mixing term and heat fluxes entering as sources and sinks. Using observational data, we apply Bayesian inference to calibrate the model. In contrast with a slab model, our model captures the exponential reduction of the diurnal warming amplitude with increasing wind speed. Further, our model performs comparably to a more elaborately parameterized diurnal warm layer model. Formulated as a single partial differential equation with three key tuning parameters, the model is suitable as an interactive numerical boundary condition for idealized atmospheric simulations.

## 1 Introduction

The role of clouds in a changing climate remains an open question that challenges predictions of climate sensitivity, regional precipitation patterns and extreme weather events (Bony et al., 2015; Siebesma et al., 2020). Fundamental processes in cloud dynamics remain insufficiently understood, particularly how convective clouds cluster and interact with their environment. This knowledge gap calls for idealized modeling approaches simple enough to distill mechanisms, yet relatable to the real world.

In the tropics, clouds organize across a wide range of spatial and temporal scales (Moncrieff, 2010). Diurnally, thunderstorms often cluster in mesoscale convective systems (MCSs), which are associated with extreme precipitation and the genesis of tropical cyclones (Tan et al., 2015; Schumacher and Rasmussen, 2020). Intraseasonally, variability is dominated by the





Madden-Julian Oscillation (MJO), an eastward-propagating zone of strong deep convective activity (Madden and Julian, 1972; Zhang, 2005). Although the MJO is known to couple to the large-scale circulation and impact weather around the globe, it remains difficult to model (Jiang et al., 2020; DeMott et al., 2015).

Due to the multiscale interaction of tropical convection, small-scale processes may be key to understanding large-scale patterns (Slingo et al., 2003). Indeed, it is increasingly acknowledged that the diurnal variability of sea surface temperature (SST) can play an important role for atmospheric dynamics (Li et al., 2001; Clayson and Chen, 2002; Bernie et al., 2008; Bellenger et al., 2010; Haerter et al., 2020). Observations draw a clear link between strong diurnal SST oscillations and a diurnal cycle of marine cumulus convection (Johnson et al., 1999). Furthermore, the diurnal cycle of SST, by enhancing air-sea

heat transfer, could help trigger the active phase of the MJO (Seo et al., 2014; Woolnough et al., 2007; Zhang, 2005; Zhao and Nasuno, 2020; Karlowska et al., 2023). Neglecting diurnal SST variations may lead to a bias on the order of $10\,\mathrm{W\,m^{-2}}$ in monthly-averaged surface heat fluxes (Weihs and Bourassa, 2014), and even stronger biases on shorter timescales. These findings emphasize the relevance of resolving diurnal air-sea interactions when modeling convective organization.

However, many idealized modeling studies of tropical convection prescribe a constant SST in space and time. A popular

modeling framework is *radiative-convective equilibrium* (RCE), where constant solar forcing is balanced by outgoing long-wave radiation above a constant, homogeneous SST (Tompkins and Craig, 1998). Under RCE, the moisture field is known to spontaneously self-organize into convective clusters separated by extended dry regions (Bretherton et al., 2005). This mechanism termed *convective self-aggregation* has been associated with real-world features such as MCS formation. Yet, self-aggregation is hampered in the realistic limit of fine horizontal model resolution, calling the realism of the mechanism into

question (Yanase et al., 2020). Recent studies, imposing a diurnal oscillation of SST, demonstrate the emergence of *diurnal self-aggregation* even at fine spatial resolution (Haerter et al., 2020; Jensen et al., 2022). Likewise, spatial variations in SST have been shown to imprint themselves on the moisture field (Müller and Hohenegger, 2020; Shamekh et al., 2020b). Since SST variability can substantially alter the spatio-temporal patterns of marine tropical convection, there is a need to incorporate higher-fidelity SST representations when investigating convective processes.

Over the past years, the modeling community studying convective organization has largely addressed the issue of an interactive SST by coupling the atmosphere to a single-layer slab with fixed heat capacity, which absorbs and re-emits heat according to parameterized surface fluxes (Hohenegger and Stevens, 2016; Shamekh et al., 2020a; Coppin and Bony, 2017; Tompkins and Semie, 2021; Wing et al., 2017). These works report overall that a responsive SST slows down or even prevents the onset of convective aggregation. However, in this paper we show that slab models are inadequate for producing realistic

diurnal SST warming: they fail to capture its wind-dependence by neglecting upper ocean mixing. Yet, by no means can wind be neglected even in strongly idealizes studies, given that cold pools, bringing gusty winds, are increasingly appreciated as an integral mechanism in convective self-organization (Tompkins, 2001; Böing, 2016; Haerter, 2019; Nissen and Haerter, 2021). To address this inconsistency, we present a simplified 1D model that significantly improves the representation of diurnal SST variability compared to a slab ocean while being adaptable and affordable for coupling to cloud-resolving atmospheric models

in an idealized setting.





SST plays a key role in governing the heat and moisture exchange between the atmosphere and ocean. Whereas observed diurnal amplitudes of surface temperature are typically largest over land, a diurnal cycle of SST is also common throughout the tropical ocean (Kawai and Wada, 2007). Under strong insolation and calm conditions, a diurnal warm layer forms during the day, raising skin SST (measured directly at the surface) by up to 3-4 K. Field studies have observed extreme diurnal warming
events of 5 K or more, though these often lie in the extra-tropics or coastal regions (Gentemann et al., 2008; Minnett, 2003; Ward, 2006). Importantly, the amplitude of diurnal warming is highly sensitive to wind speed, with observations suggesting an approximately exponential decay of diurnal warming with increasing wind speed (Gentemann et al., 2003; Börner, 2021).

To first order, the diurnal warm layer results from a competition between insolation-driven thermal stratification and wind-driven mixing. On a calm, clear day after sunrise, the incident solar radiation quickly heats up the upper ocean, creating
a stably stratified density profile. This stratification leads to suppressed vertical heat exchange, hence trapping further heat near the surface in a positive feedback which can produce strong surface warming. Later, in the early afternoon when the net surface heat flux changes sign, surface cooling causes unstable density stratification and initiates vertical mixing, resulting in a deepening of the diurnal warm layer. At night, convective mixing typically acts to "reset" the temperature profile. Furthermore, wind stress induces turbulent mixing in the water column. Known as the cool skin effect, the molecular skin layer at the sea
surface is typically a few tenths of a degree colder than the water a millimeter deeper (Donlon et al., 2002; Wong and Minnett, 2018).

Upper ocean heat transfer has been modeled on various levels of complexity over the last fifty years, ranging from fully turbulence-resolving models to simplified bulk models and empirical models (Kawai and Wada, 2007). Turbulence-resolving models make up the most realistic but also computationally expensive category (Kondo et al., 1979; Mellor and Yamada, 1982;
Large et al., 1994; Kantha and Clayson, 1994; Noh and Jin Kim, 1999; Stull and Kraus, 1987). Though such ocean models have been coupled to a weather forecasting model (Noh et al., 2011), they are arguably too costly and complex for the purpose of idealized atmospheric simulations at high horizontal resolution. The simplest models, derived from empirical relations, specialize on estimating the daily SST amplitude (Webster et al., 1996; Price et al., 1987; Kawai and Kawamura, 2002) or its hourly evolution (Li et al., 2001; Zeng et al., 1999; Gentemann et al., 2003) based on averaged atmospheric data. These
models are not designed for the high temporal resolution of atmospheric large-eddy simulations (on the order of seconds), and generally not physics-informed.

Alternatively, simplified models based on boundary layer physics have been developed in the spirit of subdividing the water column into layers described by bulk dynamical equations. A popular candidate, which we compare against here, is the prognostic ZB05 scheme by Zeng and Beljaars (2005) with improvements by Takaya et al. (2010). It computes the sea skin
temperature from an integrated mixed layer equation combined with a cool skin scheme, assuming a power-law temperature profile. Other bulk models include the PWP model (Price et al., 1986) and its developments (Fairall et al., 1996, 2003; Gentemann et al., 2009; Schiller and Godfrey, 2005). Widely applied in weather and climate modeling (Brunke et al., 2008), such models could offer a suitable balance between physical rigor and computational cost. Nonetheless, they do not necessarily cater to the needs of idealized atmospheric modeling. First, some bulk models require unknown input about the oceanic background
state. Second, the integral descriptions of heat and momentum transfer are typically based on elaborate parameterizations,



making the models less intuitive and difficult to tune in sensitivity experiments. Third, these models do not resolve the vertical temperature profile (apart from the parameterized profile in Gentemann et al. (2009)), limiting their use as an interface to sub-surface processes that might be sensitive to diurnal warming, such as microbial ecosystems (Wurl et al., 2017).

In this paper, we present an idealized model of diurnal sea surface warming that explicitly resolves temperature in the upper
few meters of the ocean while being conceptually simple and efficient. Derived from first principles as a modified heat equation, the model consists of a single partial differential equation controlled by three key parameters, taking insolation and atmospheric conditions as forcing input. After describing the model (section 2), we use a Bayesian approach to calibrate and evaluate the model based on cruise observations from the tropical Eastern Pacific (section 3). We compare the performance of our model to both a typical slab model and the ZB05 scheme (section 4). Whereas slab models fail to describe diurnal SST variability,
our model accurately reproduces key features of diurnal warming, such as its wind dependence, near-surface heat trapping, and skin cooling. Our model performs comparably to the ZB05 scheme, which is more rigorously derived but does not resolve temperature vertically. In section 4.4, we discuss the implications of our results for the atmospheric heat and moisture budget. We believe that our modular scheme can be useful both as a wind-responsive ocean surface for cloud-resolving modeling and as an interface to study further air-sea-biosphere interactions in an idealized setting.

## 2  Model description

Our approach relies on the following basic assumptions. First, we assume that the oceanic, atmospherically-driven diurnal temperature variability is constrained to within a few meters from the sea surface. We thus define the *foundation depth* $z_f$ at which diurnal temperature changes become negligible. Conceptually, this partitions the water column into the *diurnal layer* above $z_f$ and the *foundation layer* below. Second, we assume that we may separate the scales of diurnal warming from the
ocean's slow internal temperature variability. The foundation layer may then be considered as an infinite heat reservoir with constant *foundation temperature* $T_f$. Third, we neglect any horizontal inhomogeneities and flows, allowing us to treat the problem as one-dimensional. The goal is then to determine the evolution of the vertical temperature profile within the diurnal layer, in response to a given time sequence of atmospheric forcing. A schematic sketch of the setup is depicted in Fig. 1.

### 2.1  Main equation

We consider the sea temperature $T(z,t)$ as a function of time $t \geq 0$ and vertical coordinate $z \in (z_f, 0]$, where $z = 0$ defines the air-sea interface and $z < 0$ is below the surface. Given an initial profile $T(z,0)$, foundation temperature $T_f$, and atmospheric forcing $\mathcal{F}(t)$, we propose a single prognostic partial differential equation (PDE) for the time evolution of $T \equiv T(z,t)$:

$$\frac{\partial T}{\partial t} = \underbrace{\frac{\partial}{\partial z}\left(\kappa(z,t)\frac{\partial T}{\partial z}\right)}_{\text{diffusion}} - \underbrace{\mu\frac{T - T_f}{z - z_f}}_{\text{mixing}} + \underbrace{\frac{1}{\rho_w c_p}\frac{\partial Q(z,t)}{\partial z}}_{\text{source/sink}} \ . \tag{1}$$

Here $\kappa > 0$ represents the (time- and depth-dependent) diffusivity, $\mu > 0$ is a constant which we term mixing coefficient, and the constants $\rho_w$ and $c_p$ denote the density and specific heat capacity at constant pressure of sea water, respectively. Finally, $Q$





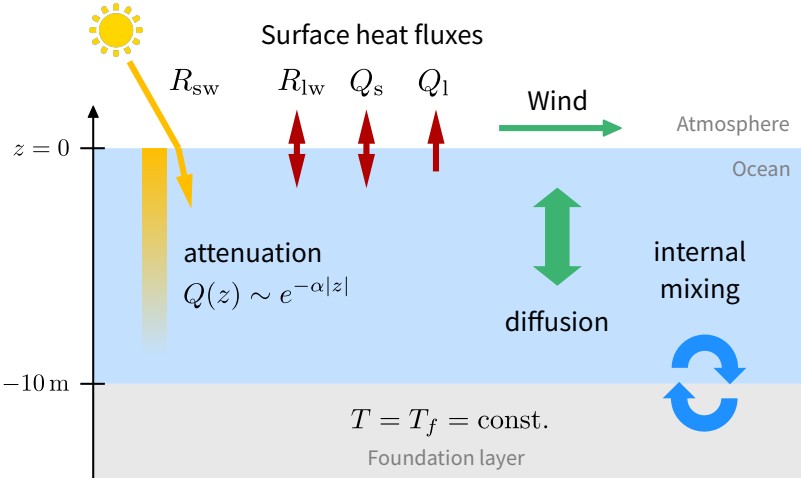

**Figure 1.** Schematic of the simplified upper ocean model, illustrating the processes considered.

is the net vertical heat flux at depth $z$, defined positive downwards (into the ocean). Explicit expressions of these quantities in relation to the forcing $\mathcal{F}$ follow below.

The three terms on the right-hand side of equation (1) provide strongly idealized representations of different physical processes. Motivated by the heat equation, the first term describes the vertical *diffusion* of heat due to turbulent eddies within the diurnal layer. In addition, internal processes such as convection may cause additional vertical *mixing* of water masses between the diurnal layer and the foundation layer. This is crudely incorporated by the second term, which relaxes $T$ to the foundation temperature with relaxation time scale $t_\mu = (z - z_f)/\mu$. Here the depth dependence reflects the intuition that water near the foundation depth will mix faster with foundation-layer water compared to water near the surface. Lastly, the third term comprises all *sources and sinks* of heat, both at the air-sea interface and within the diurnal layer.

The foundation temperature $T(z_f, t) = T_f \ \forall t$ acts as a Dirichlet boundary condition at $z_f$. At the air-sea interface ($z = 0$), the sea temperature evolves according to the net surface heat flux $Q_0(t) \equiv Q(0, t)$ (see section 2.3), which closes the energy budget.

## 2.2 Wind-driven mixing and stratification

Wind stress at the sea surface induces shear instability, causing vertical turbulent heat transport in the upper ocean. This implies that the diffusivity $\kappa(z, t)$ in our model should depend on the wind speed $u(t)$. Since the magnitude of wind stress on a water surface is approximately proportional to the square of the wind speed (Smith, 1988), we propose to model the diffusivity $\kappa$ as

$$\kappa(z, t) := \kappa_{\mathrm{mol}} + \kappa_0 \varphi(z) \left( \frac{u(t)}{u_0} \right)^2 , \tag{2}$$



where $\kappa_{\mathrm{mol}}$ and $\kappa_0$ are coefficients of molecular and (vertical) eddy diffusion, respectively. Typically, molecular heat conduction is negligible ($\kappa_{\mathrm{mol}} \ll \kappa_0$). Note the scaling of the eddy diffusion term with $u^2$, which we non-dimensionalize by the reference wind speed $u_0 = 1\,\mathrm{m\,s}^{-1}$ simply to ensure that $\kappa_0$ has units of diffusivity.

The vertical diffusivity profile $\varphi(z)$ in Eq. (2) approximates how turbulent mixing varies with depth. Physically, thermal
density stratification inhibits turbulent heat transfer, since vertical mixing of a stratified fluid is energetically unfavorable. Additionally, the characteristic size of eddies decreases when approaching the air-sea boundary (Pope and Pope, 2000). Both effects motivate suppressing the diffusivity near the surface, where stratification becomes largest during diurnal heating. For simplicity, we limit our study to a time-independent linear profile,

$$\varphi(z) = 1 + \sigma\left(\frac{z}{z_f} - 1\right) , \tag{3}$$

parameterized by the surface suppressivity $\sigma \in [0, 1]$. If $\sigma > 0$, then $\varphi$ increases linearly with depth from $\varphi(0) = 1 - \sigma$ at the
surface to $\varphi(z_f) = 1$ at the foundation depth.

It may seem counter-intuitive to decrease the diffusivity towards the surface when it is driven by surface winds. In this idealized setup, however, the diffusion term simultaneously accounts for density stratification (in a time-averaged sense). Furthermore, surface wind stress can accelerate the warm layer, leading to turbulence due to shear instability at its base (Hughes et al., 2020). Alternative choices for the diffusivity profile are discussed in section 5 and the appendix.

## 2.3 Air-sea interaction

In our model, the state of the atmosphere enters as the forcing $\mathcal{F}(t) \equiv (R_{sw,\downarrow}(t), u(t), T_a(t), q_v(t))$, comprising the downward solar, or "shortwave", irradiance $R_{sw,\downarrow}$, horizontal wind speed $u$, air temperature $T_a$, and specific humidity $q_v$. These quantities refer to some reference height above the sea surface (usually $10\,\mathrm{m}$, where they are typically measured (Friehe and Schmitt, 1976)). The diurnal layer interacts with the atmosphere through the absorption and reflection of shortwave radiation,
the absorption and emission of thermal ("longwave") radiation, as well as sensible and latent heat exchange. While shortwave radiation penetrates the sea surface and is absorbed at a range of depths, the other fluxes act within micrometers of the air-sea interface (Wong and Minnett, 2018).

At the air-sea interface, the net surface heat flux $Q_0(t)$ entering the water body is given by

$$Q_0(t) = R_{\mathrm{sw}}(t) + R_{\mathrm{lw}}(t) + Q_{\mathrm{s}}(t) + Q_{\mathrm{l}}(t) , \tag{4}$$

where $R_{\mathrm{sw}} = (1 - \mathcal{R})R_{sw,\downarrow}$ is the penetrating shortwave irradiance after subtracting from $R_{sw,\downarrow}$ the fraction $\mathcal{R}$ that is reflected
at the sea surface (see appendix). The net longwave radiative flux $R_{\mathrm{lw}}$ and sensible heat flux $Q_{\mathrm{s}}$ can point into or out of the ocean, whereas the latent heat flux $Q_{\mathrm{l}}$ is always negative. We use standard bulk formulae to describe the surface heat fluxes





(Friehe and Schmitt, 1976; DeCosmo et al., 1996; Wells and King-Hele, 1990),

$$R_{\mathrm{lw}}(t) = \sigma_{\mathrm{SB}}\left(T_a(t)^4 - T(0,t)^4\right), \tag{5a}$$

$$Q_{\mathrm{s}}(t) = \rho_a c_{p,a} C_s u(t)\left(T_a(t) - T(0,t)\right), \tag{5b}$$

$$Q_{\mathrm{l}}(t) = \rho_a C_l L_v u(t)\left(q_v(t) - q_{\mathrm{sat}}(T(0,t))\right), \tag{5c}$$

given in terms of the Stefan-Boltzmann constant $\sigma_{\mathrm{SB}}$, the density and specific heat capacity of air, $\rho_a$ and $c_{p,a}$, respectively, as well as the Stanton number $C_s$, Dalton number $C_l$, and latent heat of vaporization $L_v$. We approximate these coefficients

by constants based on literature values (see table 1), which are roughly valid for wind speeds on the order of 1 to $10\,\mathrm{m\,s^{-1}}$ and air-sea temperature differences around $1\,\mathrm{K}$, measured $10\,\mathrm{m}$ above sea level (Wells and King-Hele, 1990). Note that Eq. (5a) simply applies the Stefan-Boltzmann law for black body radiation. Furthermore, the latent heat flux, Eq. (5c), involves the saturation specific humidity $q_{\mathrm{sat}}(T)$. The temperature dependence of $q_{\mathrm{sat}}$ obeys the Clausius-Clapeyron relation, which is approximated by the empirical formula

$$q_{\mathrm{sat}}(T) \approx \frac{611.2}{\rho_a r_w T}\exp\left(\frac{17.67(T - 273.15)}{T - 29.65}\right), \tag{6}$$

where $r_w$ denotes the gas constant of water vapor and $T$ enters in units of kelvins (Alduchov and Eskridge, 1996).

We assume that the penetrating shortwave radiation $R_{\mathrm{sw}}$ is attenuated exponentially as it propagates downward through the water column,

$$Q(z,t) = R_{\mathrm{sw}}(t)\exp\left(\frac{\alpha z}{\cos\phi'(t)}\right) \qquad \text{for } z < 0, \tag{7}$$

where $\alpha > 0$ is the *attenuation coefficient* and $\phi'$ denotes the refracted solar angle. It follows from the sun's angle relative to the surface normal, $\phi \in [0, \pi/2)$, by Snell's refraction law,

$$\phi'(t) = \arcsin\left(\frac{n_a}{n_w}\sin\phi(t)\right), \tag{8}$$

where $n_w$ and $n_a$ denote the refractive indices of sea water and air, respectively.

To summarize, our idealized model is controlled by three key parameters: (i) the eddy diffusivity $\kappa_0$ governs the magnitude of wind-driven turbulent heat diffusion; (ii) the mixing coefficient $\mu$ sets the time scale of relaxation to the foundation temperature; and (iii) the attenuation coefficient $\alpha$ regulates how deep shortwave radiation penetrates. Realistic values of these parameters are estimated from observational evidence (section 3). We list all model constants and coefficients in Table 1.

## 2.4 Numerical implementation

To solve Eq. (1), we discretize the spatial coordinate $z$ and numerically integrate the resulting system of ordinary differential equations in time (see Appendix A and Börner (2024) for a software implementation in Python). Spatial derivatives are approximated by second-order accurate finite differences on a non-uniform vertical grid. We set the foundation depth to $z_f = -10\,\mathrm{m}$, where diurnal temperature variability is mostly negligible (Kawai and Wada, 2007). The grid spacing is set to $\Delta z_0 = 0.1\,\mathrm{m}$ at





| Quantity | Symbol | Unit | Value |
|---|---|---|---|
| Eddy diffusivity | $\kappa_0$ | $\mathrm{m^2\,s^{-1}}$ | $1.34 \times 10^{-4}$ * |
| Mixing coefficient | $\mu$ | $\mathrm{m\,s^{-1}}$ | $2.85 \times 10^{-3}$ * |
| Attenuation coefficient | $\alpha$ | $\mathrm{m^{-1}}$ | 3.52 * |
| Surface suppressivity | $\sigma$ | – | 0.8 |
| Foundation temperature | $T_f$ | K | 298.19 ** |
| Foundation depth | $z_f$ | m | $-10$ |
| Molecular diffusivity (water) | $\kappa_{\mathrm{mol}}$ | $\mathrm{m^2\,s^{-1}}$ | $1 \times 10^{-7}$ |
| Specific heat (water) | $c_p$ | $\mathrm{J\,K^{-1}\,kg^{-1}}$ | 3850 |
| Specific heat (air) | $c_{p,a}$ | $\mathrm{J\,K^{-1}\,kg^{-1}}$ | 1005 |
| Density (water) | $\rho_w$ | $\mathrm{kg\,m^{-3}}$ | 1027 |
| Density (air) | $\rho_a$ | $\mathrm{kg\,m^{-3}}$ | 1.1 |
| Refractive index (water) | $n_w$ | – | 1.34 |
| Refractive index (air) | $n_a$ | – | 1.00 |
| Stanton number | $C_{\mathrm{s}}$ | – | $1.3 \times 10^{-3}$ |
| Dalton number | $C_{\mathrm{l}}$ | – | $1.5 \times 10^{-3}$ |
| Latent heat of vaporization | $L$ | $\mathrm{J\,kg^{-1}}$ | $2.5 \times 10^6$ |
| Stefan-Boltzmann const. | $\sigma_{\mathrm{SB}}$ | $\mathrm{W\,m^{-2}\,K^{-4}}$ | $5.67 \times 10^{-8}$ |
| Gas constant (water vapor) | $r_w$ | $\mathrm{J\,K^{-1}\,kg^{-1}}$ | 461.51 |
| Grid spacing at surface | $\Delta z_0$ | m | 0.1 |
| Number of vertical grid points | $N$ | – | 40 |

\* determined from data via Bayesian inference, see table D1.

\*\* corresponds to mean $3\,\mathrm{m}$ water temperature of MOCE-5 data set

**Table 1.** Model constants and their default values used.

the sea surface and increases by a stretch factor $\epsilon \approx 1.04$ with each consecutive grid point below, such that a total of $N = 40$ grid points cover the diurnal layer $z \in (z_f, 0]$ (excluding the boundary point at $z_f$). Specifically, the depth $z_n$ of the $n$-th grid point is given by

$$z_n = -\Delta z_0 \left( \frac{1 - \epsilon^n}{1 - \epsilon} \right), \qquad n \in \{0, \ldots, N\}, \tag{9}$$

with $\epsilon$ set such that $z_N = z_f$.

We implement the time integration as an explicit Euler scheme, which is numerically stable if the chosen time step $\Delta t$ satisfies the Courant-Friedrichs-Lewy (CFL) condition,

$$\mathcal{C}_i := \max_n 2 \frac{\kappa(z_n, t_i) \Delta t_i}{(\Delta z_n)^2} \le 1, \qquad n = 0, 1, 2, \ldots, N - 1, \tag{10}$$

where $n$ and $i$ index the discrete space and time coordinates, respectively. Via the time-dependent diffusivity $\kappa$, the CFL condition depends on time, specifically on the current wind speed. To minimize computational cost, we use an adaptive time





step $\Delta t_i$ that maintains a maximal CFL number of $\mathcal{C}_i = 0.95$ at each instant of time $t_i$. Consequently, the time step $\Delta t_i$ scales inversely with the square of the wind speed. We impose a cut-off wind speed $u_{\max} = 10\,\mathrm{m\,s^{-1}}$, causing any wind speed $u > u_{\max}$

to be replaced by $u_{\max}$ when computing the diffusivity (eq. (2)).

Comparing the explicit Euler scheme with analogous implementations of the fourth-order Runge-Kutta and implicit Euler methods, we find that the explicit Euler scheme is fastest unless the implicit method is used at very large time step, in which case the solution lacks accuracy.

## 3  Model calibration and evaluation

Our model parameterizes diurnal warming in terms of three unknown constants: the diffusivity coefficient $\kappa_0$, mixing coefficient $\mu$, and attenuation coefficient $\alpha$. In principle, their values depend on the ocean properties at a given time and location. In this section, we first analyze how the parameters affect diurnal warming, and then use observational data to estimate realistic values via Bayesian inference. Finally, we evaluate the performance of the calibrated model against observations.

### 3.1  Parameter sensitivity under idealized forcing

As an initial step, we perform a sensitivity study where we force the model with idealized atmospheric pseudo-data representing a calm and clear tropical day. Setting the foundation temperature to $T_f = 300\,\mathrm{K}$, we let the air temperature $T_a(t)$ oscillate harmonically around $T_f$ with a diurnal amplitude of $\Delta T = 2\,\mathrm{K}$. Similarly, we impose a harmonically oscillating horizontal wind speed with a mean of $\overline{u} = 2\,\mathrm{m\,s^{-1}}$ and amplitude $\Delta u = 2\,\mathrm{m\,s^{-1}}$, peaking at midnight (this choice generates favorable conditions for diurnal warming and nighttime mixing). Such profiles read:

$$T_a(t) = T_f - \frac{\Delta T}{2}\cos(2\pi t/t_0)\,; \qquad u(t) = \overline{u} + \frac{\Delta u}{2}\cos(2\pi t/t_0)\,, \tag{11}$$

where $t$ is local sun time (with respect to midnight) and $t_0 = 1\,\mathrm{d}$. The solar irradiance is given by

$$R_{\mathrm{sw},\downarrow}(t) = \begin{cases} -R_{\max}\cos(2\pi t/t_0) & \text{if } \cos(2\pi t/t_0) < 0 \\ 0 & \text{otherwise,} \end{cases} \tag{12}$$

where we set the peak insolation $R_{\max} = 1000\,\mathrm{W\,m^{-2}}$. Lastly, we fix the specific humidity at a constant value of $q_v = 15\,\mathrm{g\,kg^{-1}}$. Under these conditions, we may expect peak diurnal surface warming of around 2-3 K (Minnett, 2003).

Using the atmospheric pseudo-data described above, we now run two-day-long model simulations, consecutively varying one model parameter of the set $\{\kappa_0, \mu, \alpha\}$ while fixing the other two at select default values (see Fig. 2). Increasing the eddy

diffusivity $\kappa_0$ enhances the heat transport within the diurnal layer, flattening the vertical temperature gradient, diminishing surface warming and advancing the time of peak surface warming. A decrease in the mixing coefficient $\mu$ corresponds to a slower removal of heat from the diurnal layer into the deeper ocean. This causes increased heat trapping in the upper ocean, hence stronger surface warming and a deeper warm layer. Particularly, for $\mu < 10^{-3}\,\mathrm{m\,s^{-1}}$, excess heat remains in the diurnal layer throughout the night, accumulating heat on the next day. Setting $\mu$ to a sufficiently large value ensures that the temperature



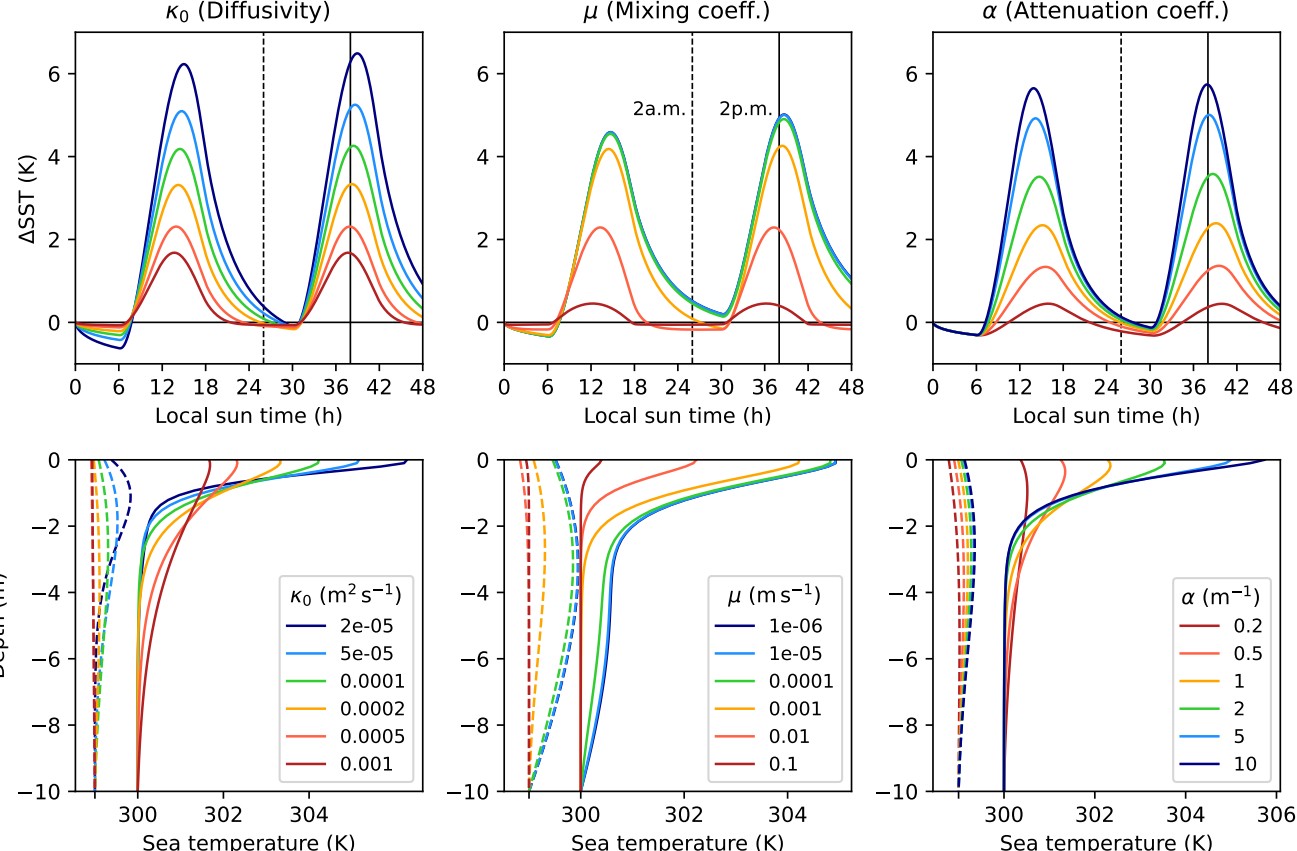

**Figure 2.** Model sensitivity under variation of the parameters $\kappa_0$, $\mu$, and $\alpha$. In each sensitivity experiment, we force the model with two days of idealized atmospheric data (see main text), successively varying one parameter while fixing the other two. The top panels depict simulated surface warming $\Delta$SST as a function of time for different values of the eddy diffusivity (left), mixing coefficient (center), and attenuation coefficient (right). The bottom panels show corresponding vertical temperature profiles at 2 a.m. (dashed lines, shifted by $-1$ K) and 2 p.m. (solid lines) on the second day of simulation. Values of the varied parameter are given in the respective legend; fixed parameter values are $\kappa_0 = 1 \times 10^{-4}\,\mathrm{m^2\,s^{-1}}$, $\mu = 1 \times 10^{-3}\,\mathrm{m\,s^{-1}}$, and $\alpha = 3\,\mathrm{m^{-1}}$. Further model settings are detailed in table 1 (here $T_f = 300\,\mathrm{K}$).





profile can "reset" at night, as is often observed (Kawai and Wada, 2007). Finally, the attenuation coefficient $\alpha$ determines the depth range in which solar radiation is absorbed. For $\alpha > 1\,\mathrm{m}^{-1}$, more than 60% of radiation is absorbed within $1\,\mathrm{m}$ of the surface, leading to strong surface warming. Reducing $\alpha$ causes radiation to reach deeper, where it is more quickly transported into the foundation ocean via the mixing term.

In Fig. 2, the similarity in model response between the parameters $\kappa_0$ and $\alpha$ suggests that they might play a similar dynamical

role. However, this is not the case. The two parameters control different aspects of the air-sea coupling. The eddy diffusivity $\kappa_0$ determines the coupling to wind, whereas the attenuation coefficient $\alpha$ modulates the oceanic heat uptake due to solar radiation. Correlations between model parameters are discussed below (section 3.3).

## 3.2   Observational data

To infer realistic values for the parameter set $\{\kappa_0, \mu, \alpha\}$, we now conduct a case study where we force the model with real

observational data and compare the modeled diurnal warming with the observed signal. Here we use cruise data (available from Börner (2024)) from the Fifth Marine Optical Characterization Experiment (MOCE-5), conducted during October 1999 off the Mexican west coast (Minnett, 2003; Ward, 2006). The route led along the coast of the Baja California peninsula, both in the open Pacific Ocean and within the Gulf of California, thus including offshore as well as more coastal conditions (see Fig. 3a).

The research vessel *Melville* was equipped with an infrared radiometer of type M-AERI (Marine-Atmospheric Emitted Radiance Interferometer, Minnett et al. 2001). This instrument provides precise measurements of sea skin temperature by detecting infrared radiation emitted from within micrometers of the ocean surface. Using skin SST for calibration, rather than bulk SST measured at up to $1\,\mathrm{m}$ depth, is crucial when modeling air-sea interactions because the atmosphere senses only the sea skin. Additionally, solar irradiance $R_{\mathrm{sw},\downarrow}$, wind speed $u$, air temperature $T_a$, and water temperature at $3\,\mathrm{m}$ depth were recorded

at time intervals of approximately 10 to 12 minutes (see Fig. 3). Unfortunately, the data set does not contain air humidity. We therefore assume a constant specific humidity of $q_v = 15\,\mathrm{g\,kg}^{-1}$ throughout the time series, which may be considered typical for the tropical ocean.

Throughout the following analysis, following Minnett (2003), we define *diurnal warming* $\Delta$SST as the temperature difference between the sea skin (that is, the water directly at the surface) and a reference depth $d$,

$$\Delta\mathrm{SST}(t) := T(0,t) - T(d,t)\,,$$

where $d = -3\,\mathrm{m}$ for the present data set and $T(0,t)$ is given by the radiometric SST measurements. Diurnal warming events exceeding $1\,^\circ\mathrm{C}$ are observed on several days, with $\Delta$SST reaching up to 5°C on Oct 13 (Fig. 3c; see also Ward (2006)). The data set also includes days without any substantial diurnal warming, such as on Oct 2 and Oct 9. These days correlate with relatively high wind speeds, while the strong warming events in the second week, such as on Oct 10, 13, and 14, coincide with

low winds especially during midday. Note that the time series includes diurnal warming events at different locations (Fig. 3b) and covers a wide range of SST values from 290 to $305\,\mathrm{K}$ (Fig. 3d). Horizontal wind speeds (corrected for the ship's motion) did not exceed $10\,\mathrm{m\,s}^{-1}$; downward solar irradiance peaked at around 800 to $900\,\mathrm{W\,m}^{-2}$ each day.





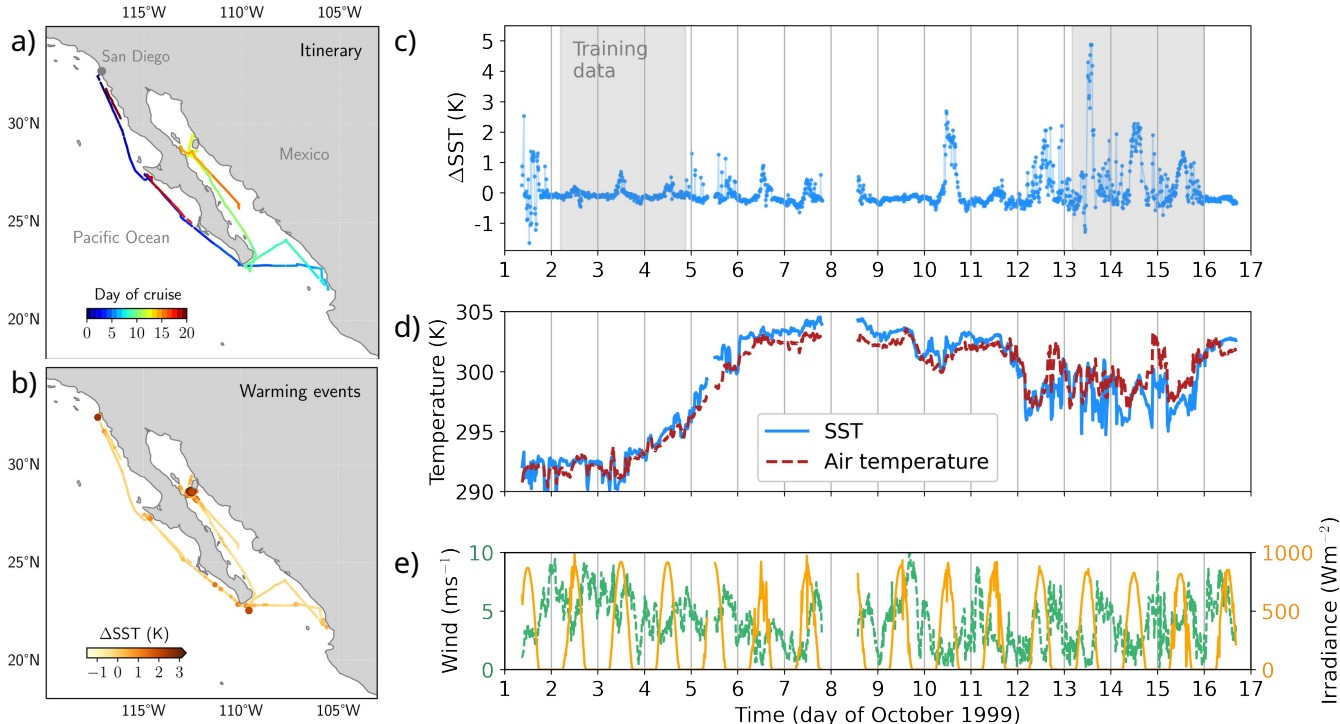

**Figure 3.** Observational data set used in this study. a) Travel route of the MOCE-5 cruise in the Pacific Ocean and Gulf of California, colored by the day since departure near San Diego, USA. b) Diurnal warming $\Delta$SST by location, as recorded during the cruise (data points with higher $\Delta$SST are enlarged). c) Time series of observed diurnal warming, $\Delta$SST, showing the individual data points. Gray shaded intervals indicate the training data used for Bayesian inference. Panels d) and e) display the time series of radiometric skin SST (blue), air temperature (red), horizontal wind speed (green), and downward shortwave irradiance (orange). Note that we omit data recorded after Oct 16 due to extended temporal gaps in the data set.

While the MOCE-5 cruise data has a temporal resolution of around 10 minutes, our model requires a time step on the order of seconds to meet the CFL condition (eq. (10)). This necessitates interpolating the atmospheric data between data points; we
apply linear interpolation with a wind-adaptive time step as described in section 2.4.

### 3.3 Bayesian parameter estimation

The MOCE-5 observations allow us to estimate the free parameters of our model using Bayesian inference. Given the data $\mathcal{D}$, the conditional probability $P(\Theta|\mathcal{D})$ that a certain value of the parameter set $\Theta = \{\kappa_0, \mu, \alpha\}$ represents the "true" model follows from Bayes' rule,

$$P(\Theta|\mathcal{D}) \propto P(\mathcal{D}|\Theta)P(\Theta) \,, \qquad\qquad (13)$$





that is, the *posterior* distribution $P(\Theta|\mathcal{D})$ is proportional to the product of the *likelihood* $P(\mathcal{D}|\Theta)$ and the *prior* distribution $P(\Theta)$. The likelihood quantifies how well the model with given parameter settings $\Theta$ describes the observed data, while the prior distribution encapsulates previous knowledge of suitable parameter ranges (Gelman et al., 2013).

To estimate the posterior distribution, we first partition the data time series into a six-day *training* set and an about eight-day *validation* set (see gray shading in Fig. 3c). The training data are used to evaluate the likelihood function $\mathcal{L}(\Theta) \equiv P(\mathcal{D}|\Theta)$, which is computed in the following way. For given $\Theta$, the model is run using the time series of the variables $\mathcal{D} = \{R_{\mathrm{sw},\downarrow}, u, T_a, q_v\}$ as atmospheric forcing. We set the foundation temperature $T_f$ to the mean observed sea temperature at $3\,\mathrm{m}$ depth, $T_f = 298.19\,\mathrm{K}$, and shift the air temperature $T_a$ according to the current deviation of the $3\,\mathrm{m}$-temperature from its mean $T_f$. This maintains the observed air-sea temperature contrast at all times. Then, we compare the modeled time series of diurnal warming, $\Delta\mathrm{SST}^{\mathrm{model}}$, to the observed diurnal warming signal $\Delta\mathrm{SST}^{\mathrm{obs}}$ in order to compute the likelihood,

$$\mathcal{L}(\Theta) \propto \exp\left(-\sum_j \frac{\left(\Delta\mathrm{SST}_j^{\mathrm{model}}(\Theta) - \Delta\mathrm{SST}_j^{\mathrm{obs}}\right)^2}{\Sigma_j^2}\right). \tag{14}$$

Here the index $j$ runs through all data points (over time) in the training set, and $\Sigma_j$ denotes a weight (see appendix).

Approximating the posterior distribution in parameter space is achieved via Markov chain Monte Carlo (MCMC) sampling using the `emcee` package in Python (Foreman-Mackey et al., 2013), based on an affine-invariant algorithm proposed by Goodman and Weare (2010). Further information on the Bayesian inference procedure and choice of prior is provided in the appendix.

After convergence, the sampled posterior distribution shows a single peak in each parameter direction, indicating a well-defined optimal value for each parameter (Fig. 4). From the projected distributions, we compute the maximum, median, and mean value as parameter estimates (tab. D1). The maximum value, or *maximum a posteriori* (MAP) estimate, corresponds to the most likely parameter value and will be used for the subsequent analysis.

The two-dimensional projections of the posterior distribution (Fig. 4) indicate possible correlations between the three parameters. In particular, a positive correlation between $\kappa_0$ and $\alpha$ is visible. However, part of the correlation likely reflects the fact that the solar irradiance (modulated by $\alpha$) and wind speed (modulated by $\kappa_0$) are themselves negatively correlated in the training data (not shown).

## 3.4 Model performance against observations

After estimating the parameters $\kappa_0$, $\mu$, and $\alpha$ via Bayesian inference, we now evaluate the performance of the calibrated model against the full observational data set, using the settings given in table 1. We force the model with the observed 16-day time series of solar irradiance, wind speed, and air temperature, interpolated in time to match the CFL condition (Eq. (10)). As in the previous section, we set the foundation temperature to the observed mean $T_f = 298.19\,\mathrm{K}$ and adjust the air temperature time series accordingly.

Fig. 5 illustrates the results of this simulation in comparison with the observed diurnal warming. Similar to the observations, the model produces a wide range of diurnal warming amplitudes from $0.2\,\mathrm{K}$ on day 9 to almost $4\,\mathrm{K}$ on day 13, which coin-



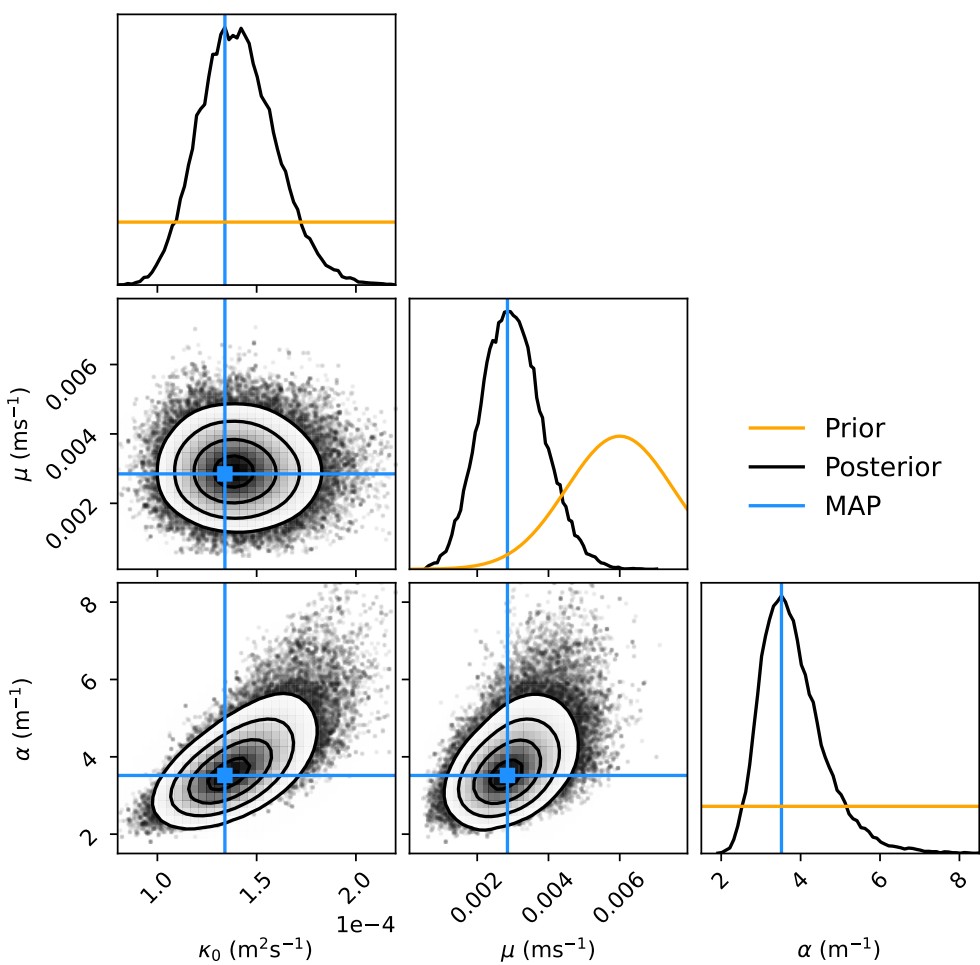

**Figure 4.** Posterior distribution of the parameter set $\{\kappa_0, \mu, \alpha\}$, as obtained from Bayesian MCMC sampling. Plots on the upper diagonal are 1D projections of the parameter space, showing for each parameter the prior (orange) and sampled posterior (black) distribution (vertical axis not to scale). Blue lines indicate the MAP values (table D1). The 2D projections of parameter space depict scatter points and smoothed iso-contours of the posterior. Plotted using corner.py (Foreman-Mackey, 2016).



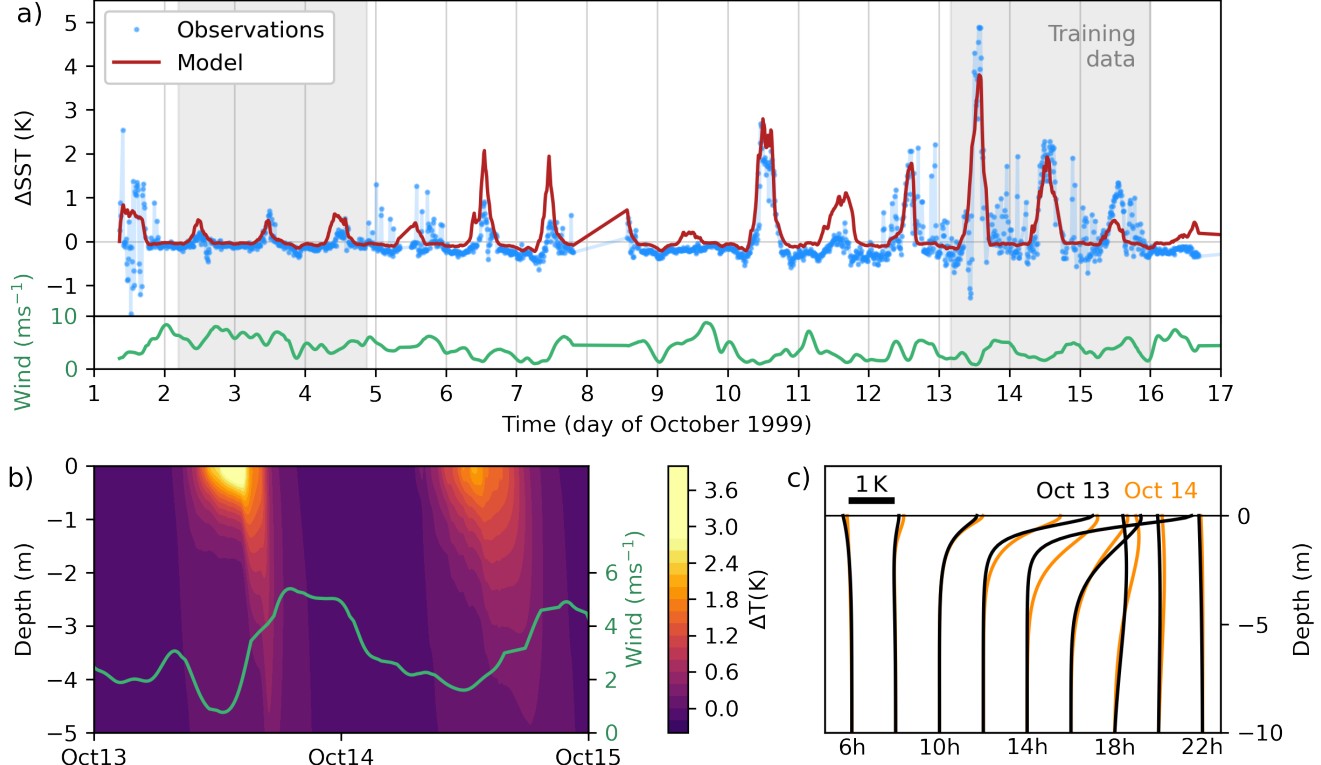

**Figure 5.** Simulation results for the calibrated model, forced with the observational data set. a) Modeled (red) and observed (blue) time series of diurnal warming. Grey shaded intervals indicate the training data used for calibration. The subpanel below depicts wind speed (green, smoothed) for reference. b) Modeled sea temperature $\Delta T = T - T_f$ as a function of depth and time, shown for day 13, 00.00h to day 15, 00.00h (local sun time). The green line indicates wind speed. c) Modeled vertical temperature profiles on day 13 (black) and 14 (orange), plotted at intervals of two hours (shifted by $0.5\,\mathrm{K}$ per hour). The scale bar indicates a temperature difference of $1\,\mathrm{K}$ along the x-axis.

cides with the maximal observed diurnal warming of $4.9\,\mathrm{K}$ (Fig. 5a). Modeled diurnal warming peaks align in time with the observations. On days 6 and 7, the model overestimates $\Delta$SST by more than $1\,\mathrm{K}$ (see discussion, section 5). The variation in

diurnal warming amplitudes links closely to wind speed (Fig. 7b). To some extent, the model captures skin cooling at night, where $\Delta$SST drops below $0\,\mathrm{K}$ in agreement with the observations (e.g. on the nights of days 3-4, 6-7, 9-10, and 12-13, see Fig. 5a). Overall, the reduction of $\Delta$SST due to skin cooling is less by about $0.07\,\mathrm{K}$ in the model than in observations.

      For the whole time series, the modeled and observed $\Delta$SST are correlated with a Pearson correlation coefficient of 0.74 (Fig. 6b). Considering only the training data, the correlation coefficient is 0.82, whereas the value for all data points outside of the

training data is 0.67. This evidences that our model has predictive skill in situations which the model has not been calibrated to. In particular, the model predicts both the absence of diurnal warming on day 9 and the strong consecutive warming event on day 10 (Fig. 5a). The model performance is further discussed in comparison with other models (sections 4 and 5).





## 3.5 Depth-resolved temperature profile

Rather than simulating diurnal warming only at the surface, our model provides the vertical temperature profile within the
upper $10\,\mathrm{m}$ of the ocean (Figs. 5b, c). In agreement with observations, heat trapping under calm and clear conditions occurs in
the uppermost meter (Soloviev and Lukas, 1997; Ward, 2006). Following peak warming, the model exhibits a deepening of the
warm layer between noon and sunset, as seen in Price et al. (1986) and Soloviev and Lukas (1997).

Analyzing the temporal evolution of the temperature profile on days 13 and 14 reveals the close connection to the wind
(Fig. 5b), with increased winds causing a fast diffusion of the warm layer on day 13 after noon and overall weaker temperature
gradients on day 14. Night-time skin cooling is visible as a slightly negative near-surface temperature gradient between around
22.00h and 06.00h (Fig. 5c). The fact that the model produces qualitatively realistic temperature profiles, even though its
parameters were merely calibrated with respect to the temperature difference between the surface and a reference depth,
supports the physical basis of the conceptual model.

## 4 Comparison with other models

Our model (hereafter referred to as *DiuSST*) is intended for use as an interactive SST boundary in idealized atmospheric
simulations. This calls for a comparison with existing models that could be selected for this purpose. First, we consider a slab
ocean model of the type previously used in atmospheric convection studies (Hohenegger and Stevens, 2016; Shamekh et al.,
2020a; Coppin and Bony, 2017; Tompkins and Semie, 2021). Second, we test the prognostic scheme by Zeng and Beljaars
(2005), which has been widely applied in weather and climate modeling but, to the best of our knowledge, not been employed
by the idealized convection community. Finally, we compare all models against the observations of the MOCE-5 data set and
discuss their effect on air-sea heat exchange.

### 4.1 Slab model

We use a single-layer slab with temperature dynamics described by

$$\dot{T}(t) = \frac{Q_0(t) - S}{\rho_w c_p h} - \xi_1 (T(t) - T_f) - \xi_2 \int\limits_0^t (T(t') - T_f)\,\mathrm{d}t' , \tag{15}$$

where $T$ denotes the slab temperature relative to $T_f$, $h$ is the slab thickness, $S$ represents a constant heat sink, and the net
surface heat flux $Q_0$ is given by eq. (4). The constants $\rho_w$, $c_p$, and $T_f$ are listed in table 1. In addition to the heat sink $S$,
we include two correction terms that are sometimes added to control the slab temperature: a linear relaxation and an integral
correction to prevent temperature drift. Their strengths are tuned via the parameters $\xi_1$ and $\xi_2$, respectively. Here $t > 0$ and
$t = 0$ is the time of the initial condition.

To compare this slab model with our model, we calibrate the parameter set $\Theta_{\mathrm{slab}} = \{h, S, \xi_1, \xi_2\}$ using Bayesian inference,
taking the same data and settings as when calibrating our model (see section 3.3). In the case of the slab, diurnal warming
$\Delta$SST directly corresponds to the slab temperature anomaly $T$. The resulting parameter estimates are given in table D1. The





Bayesian MCMC sampling converges to $S = 92\,\mathrm{W\,m^{-2}}$, a realistic approximation of net oceanic heat uptake in the tropics.
Simulation results are discussed together with the other models (section 4.3).

### 4.2 ZB05 model

The ZB05 model by Zeng and Beljaars (2005) consists of two components: a bulk equation describing the temperature evolution in the diurnal warm layer of depth $d$, and a skin layer equation representing the cool skin effect, that is, cooling due to surface heat fluxes within the upper millimeter of the ocean (Wong and Minnett, 2018; Fairall et al., 1996). The sea skin temperature relative to the foundation temperature $T_d$ is thus given by the sum of warm layer heating $\Delta T$ and skin cooling $\delta T$,

$$\Delta\mathrm{SST}(t) = \Delta T(t) + \delta T(t) \,.$$

In order to integrate over the warm layer, the model assumes a power-law temperature profile, parameterized by an empirical shape parameter $\nu = 0.3$. The integrated equation reads (Eq. (11) in Zeng and Beljaars (2005)):

$$\frac{\mathrm{d}(\Delta T)}{\mathrm{d}t} = \frac{Q_0 - R(-d)}{d\rho_w c_p \nu/(\nu+1)} - \frac{(\nu+1)ku_*\Delta T}{d\phi_t(d/L)} \,, \tag{16}$$

where $\Delta T = T_{-\delta} - T_d$ denotes the temperature difference between the skin layer depth $\delta$ and the reference depth $d = 3\,\mathrm{m}$, $Q_0$ is the net surface heat flux (Eq. (4)), $R(-d)$ is the downward-penetrating shortwave radiation at depth $d$, and $k = 0.4$ is the
von Karman constant. Furthermore, $u_*$ represents the friction velocity which we compute in terms of the wind speed $u$ and drag coefficient $C_D = 1.3 \times 10^{-3}$ according to $u_* = \sqrt{\rho_a C_D u^2/\rho_w}$ (Trenberth et al., 1989; Kara et al., 2007). The stability function $\phi_t$ is derived from Monin-Obukhov similarity theory and depends on the Monin-Obukhov length $L$. Here we use the refined formulation of Takaya et al. (2010) for $\phi_t$.

In describing the cool skin layer we follow the scheme by Fairall et al. (1996), as implemented in the noa (2023). The
evolution of the skin layer thickness $\delta$ and skin layer temperature difference $T_d$ are computed based on each other in an iterative fashion. The reason why we do not directly use Eq. (6) in Zeng and Beljaars (2005) to determine $\delta$ is that this equation differs from the referenced version in Fairall et al. (1996) and leads to numerical instabilities.

### 4.3 Comparison of model performance

Overall, we find that both our DiuSST model and the ZB05 model give a similar, adequate performance when tested on the
MOCE-5 dataset (Figs. 6, 7, 8). On the contrary, the slab model fails to describe the observed diurnal SST variability, even though it has been optimized on the training set of this data.

Specifically, the slab model is unable to capture the wind-dependent variation in diurnal warming amplitudes (Fig. 6a). The slab simulation exhibits a rather invariant diurnal amplitude of just above $\Delta\mathrm{SST} = 1\,\mathrm{K}$, largely independent of wind speed (Fig. 7b). The poor agreement between the slab and the observations manifests itself in the relatively low Pearson correlation
coefficient of 0.50 and a standard mean error of $0.36\,\mathrm{K}$ (over the whole data set). In comparison, the DiuSST and ZB05 models yield a Pearson correlation of 0.74 and 0.72, respectively, each deviating from the observations by around $0.29\,\mathrm{K}$ on average. Even more than DiuSST, the ZB05 model overpredicts diurnal warming on days 6 and 7. ZB05 exhibits more high-frequency





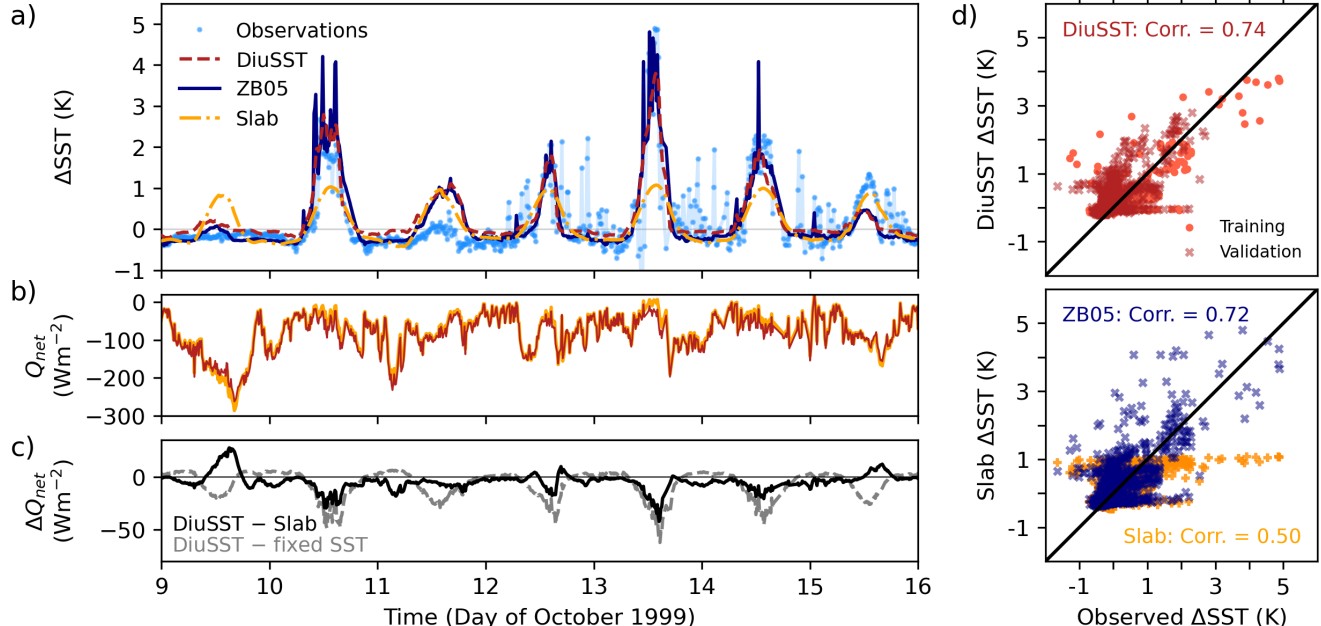

**Figure 6.** Comparison of the calibrated DiuSST, ZB05, and Slab models. a) Time series of diurnal warming $\Delta$SST as modeled by DiuSST (red), ZB05 (dark blue) and the slab (orange) for a select time interval. Points (light blue) indicate the observations. b) Net surface heat loss to the atmosphere, $Q_{\text{net}} = R_{\text{lw}} + Q_{\text{l}} + Q_{\text{s}}$ (excluding shortwave radiation), for the model (red) and slab (orange). c) Difference in $Q_{\text{net}}$ between our model and the slab (black) as well as our model and a fixed SST at $T_f$ (gray). d) Correlations between the modeled and observed $\Delta$SST for our model (top) as well as ZB05 and the slab (bottom). The Pearson correlation coefficients (Corr.) are given. In the top panel, circles (crosses) mark training (validation) data points.

variability during diurnal warming events compared to DiuSST, which we attribute to the more sensitive skin layer model explicitly included in ZB05. Night-time skin cooling is overall stronger in ZB05 relative to DiuSST, agreeing better with

observations during the second half of the time series (days 9-16) but exaggerating skin cooling during the first, windier half (days 1-7, see Fig. F1). While capturing some of the observed night-time dynamics of $\Delta$SST, the Slab tends to produce excess skin cooling especially during strong winds.

  Fig. 7b) highlights the improved wind dependence of $\Delta$SST in DiuSST compared to the Slab model. Similar to observations and ZB05, our model exhibits a roughly exponential decay of peak diurnal warming with increasing wind speed. To provide a

more quantitative analysis, we bin each diurnal warming time series by wind speed, selecting a bin size of $0.5\,\text{m s}^{-1}$. For each bin, we calculate hourly averages of $\Delta$SST as a function of local sun time and take the maximum of these hourly averages as an estimate of the mean diurnal warming amplitude for the given wind speed (Fig. 8). Based on exponential fits, $\Delta$SST as a function of wind speed decays with a scaling constant of roughly $2\,\text{m s}^{-1}$ for the observations, DiuSST and ZB05 models (Tab. 2). In the case of the Slab, the exponent is almost $30\,\text{m s}^{-1}$, confirming the weak wind sensitivity of the diurnal amplitude. The





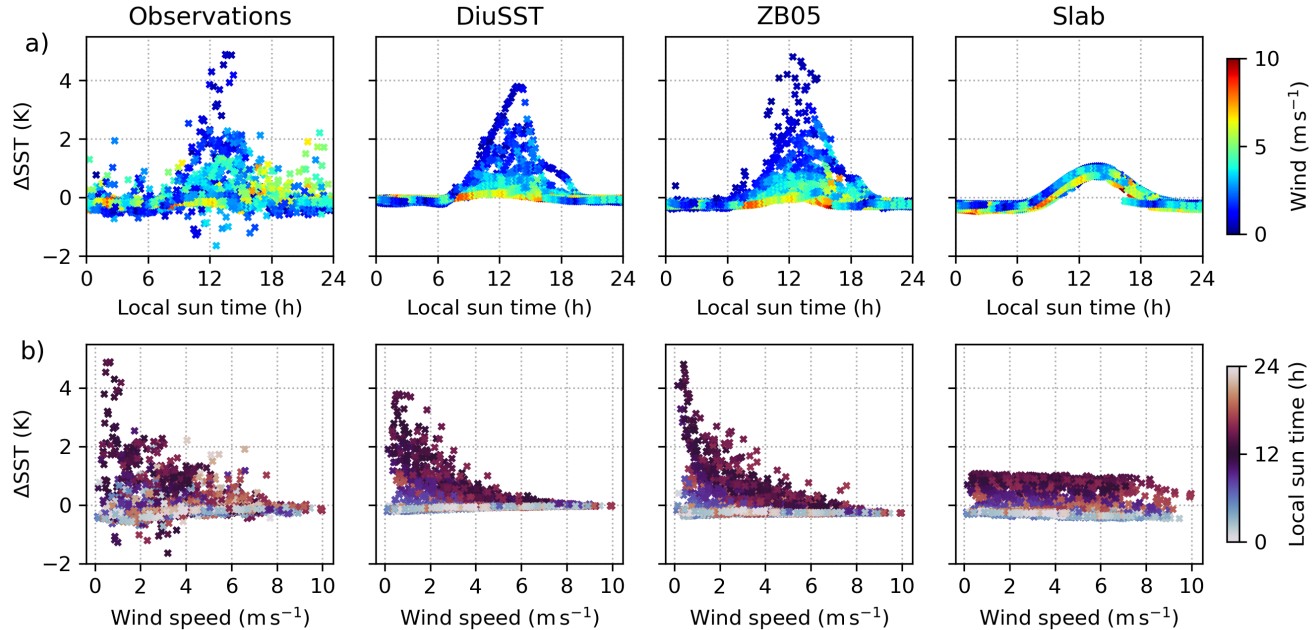

**Figure 7.** Diurnal warming in the observations and simulations with the DiuSST, ZB05, and Slab models (from left to right), showing all data points of the 16-day time series. a) $\Delta$SST as a function of local sun time, colored by wind speed. b) $\Delta$SST as a function of wind speed, colored by local sun time.

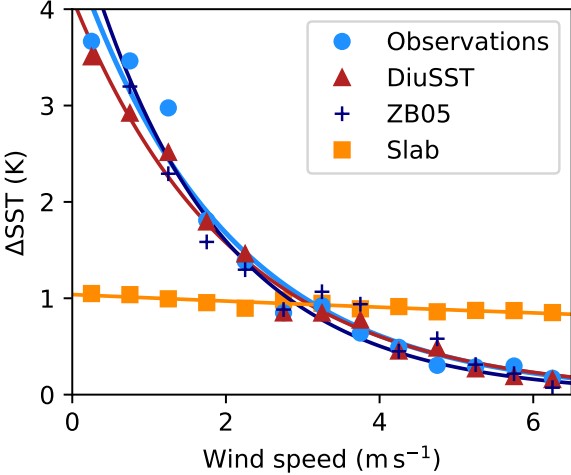

**Figure 8.** Wind dependence of diurnal warming, comparing the observations against the predictions by the DiuSST, ZB05, and Slab models (see figure legend). The data points represent the mean diurnal warming amplitude (see main text). Solid lines indicate least-squares exponential fits, with parameters given in tab. 2. Wind speeds above $6.5\,\mathrm{m\,s^{-1}}$ are not shown due to a small number of raw data points.



| Data | intercept $y_0$ (K) | exponent $a$ $(\mathrm{m\,s^{-1}})$ |
|---|---|---|
| Observations | 4.59 | 1.98 |
| DiuSST | 4.12 | 2.08 |
| ZB05 | 5.04 | 1.74 |
| Slab | 1.04 | 29.63 |

Fit function: $\Delta\mathrm{SST}(u) = y_0 \exp(-u/a)$

**Table 2.** Exponential fits of the wind dependence of diurnal warming (see Fig. 8).

expected diurnal warming amplitude under calm conditions, approximated by the intercept of the fit at $u = 0$, is underestimated (overestimated) by DiuSST (ZB05) by about $0.5\,\mathrm{K}$ each. The Slab diurnal warming amplitude is significantly too low.

### 4.4 Air-sea heat fluxes

The diurnal evolution of SST impacts the atmosphere by regulating the heat and moisture transfer at the air-sea interface. Over the course of the observational data interval (October 1-16), we compute the net surface heat flux $Q_{\mathrm{net}}(t) = R_{\mathrm{lw}}(t) + Q_{\mathrm{l}}(t) +$
$Q_{\mathrm{s}}(t)$ (excluding shortwave radiation) for each model simulation (Fig. 6b, $Q_{\mathrm{net}} < 0$ corresponds to net heat export from ocean to atmosphere). In addition, we compute $Q_{\mathrm{net}}(t)$ for the given atmospheric forcing under the assumption that SST is fixed at $T(0,t) = T_f$ for all $t$.

The oceanic heat loss to the atmosphere can differ by up to around $50\,\mathrm{W\,m^{-2}}$ between DiuSST and the Slab (Fig. 6c). $Q_{\mathrm{net}}$ is more negative for DiuSST when it produces a larger sea skin temperature compared to the Slab, and vice versa. During strong
diurnal warming events, the difference $\Delta Q_{\mathrm{net}}$ is even larger when comparing DiuSST to a fixed SST, exceeding $50\,\mathrm{W\,m^{-2}}$ on Oct 13. We conclude that using a fixed SST as an oceanic boundary condition leads to biases in surface heat fluxes particularly during strong diurnal warming events, whereas using a slab model causes biases under strong insolation both in windy and calm conditions. Between DiuSST and ZB05, differences in the net surface heat flux are relatively small and mainly due to differences in the magnitude of skin cooling.

## 5 Discussion

This study aims at contributing a diurnal warm layer model that offers a simple improvement over slab ocean models previously used in idealized atmosphere-ocean studies. Upper ocean heat transfer involves complex processes from wave breaking and Langmuir circulations to biological productivity (Edson et al., 2007; Noh et al., 2004). To first order, however, the diurnal variability of SST is governed by the competing effects of solar absorption and wind-driven turbulent mixing. This permits our
reductionist modeling approach, which approximates the temperature dynamics in the diurnal warm layer mainly as a function of wind speed and insolation. For detailed realism of diurnal warm layer dynamics, more comprehensive models (as listed in the introduction) will be appropriate.



## 5.1 Model limitations and extensions

Due to its one-dimensional setup, DiuSST neglects horizontal flows and heat exchange. Furthermore, turbulence is dissipated
instantaneously, i.e. there is no memory of momentum due to the wind stress history. These restrictions could explain the poor
performance of DiuSST (and ZB05, likewise) on days 6 and 7, when the amplitude of diurnal warming is overestimated by
more than $1\,\mathrm{K}$ (Fig. 5a). Despite strong insolation and low winds before noon on these days, observed diurnal warming does
not exceed $1\,\mathrm{K}$. This suggests the presence of enhanced turbulent vertical mixing not captured by the models, possibly due
to non-local effects such as horizontal currents or rough seas, either advected from a windy region or remnant from a windy
episode preceding the observations.

For simplicity, DiuSST represents stratification in the diurnal warm layer only in a time-averaged sense via the diffusivity
profile $\varphi(z)$ which suppresses turbulent diffusion near the surface. In reality, upper-ocean water column stability depends on the
interplay between time- and depth-dependent density gradients and turbulence dissipation (Hughes et al., 2020). Instead of the
highly simplified linear profile given in eq. (3), a nonlinear, dynamic diffusivity profile could incorporate the nature of upper
ocean turbulence more realistically. For instance, a state-dependent profile, $\varphi = \varphi(z, T(z, t))$, could reflect the temperature
dependence of stratification. We discuss alternative diffusivity profiles in C.

The mixing term controlled by $\mu$ ensures that the diurnal layer temperature relaxes back to the foundation temperature,
resetting the temperature profile at night. Combined with the diffusion term, it can only serve as a crude account of the complex
mixing processes in the upper ocean (Hughes et al., 2020), such as convective overturning or internal waves. However, the
contribution from this term is typically small compared to the other terms, particularly near the surface. In some cases, it may
be realistic to further decrease $\mu$ to allow for multi-day warming events (Jia et al., 2023).

With a vertical resolution set to $10\,\mathrm{cm}$ at the surface, our numerical implementation is too coarse to explicitly resolve the
cool skin layer, which forms within millimeters from the air-sea interface. Nonetheless, the model still captures skin cooling at
night, indicated by slightly negative values of $\Delta$SST, in agreement with the observations (Fig. 5). In fact, the model considers
the cool skin effect in a coarse-grained sense, averaging heat fluxes over the upper $10\,\mathrm{cm}$ of the water column. Of course, the
grid resolution could be increased, albeit at the cost of a smaller integration time step to maintain numerical stability.

For future refinement, further dynamical processes could be built into DiuSST via the three key modular terms. A time-
dependent diffusivity profile would allow to parameterize the effect of precipitation, which we presently neglect in the model.
Rain freshwater pools can either enhance heat trapping or mixing, depending on the competing effects of salinity and temper-
ature (Webster et al., 1996; Bellenger et al., 2017; Witte et al., 2023). As another example, a dynamic attenuation coefficient
$\alpha(t)$ would allow to account for changes in the seawater optical properties, e.g. due to microbial activity (Wurl et al., 2017).

## 5.2 Data-driven parameter estimation

While the parameters $\{\kappa_0, \alpha, \mu\}$ are fixed in our model, they generally depend on oceanic conditions that may vary in space
and time. A major benefit of Bayesian inference is that the model parameters can readily be re-calibrated to additional data of
interest.





The MOCE-5 cruise data used here cover measurements across approximately ten degrees latitude and longitude, both in the open Pacific Ocean and in coastal waters of the Gulf of California, where dynamical and optical properties of sea water are likely to have differed. The observed foundation temperature varied by several Kelvin during the cruise, whereas in the model we fix $T_f$ to the observed mean. Thus, spatial heterogeneity of water properties probably constitutes a main error source

between model and observations. For example, on day 13, the modeled maximum of diurnal warming is about $1\,\mathrm{K}$ below the observed value of $\Delta\mathrm{SST} \approx 4.9\,\mathrm{K}$. This difference may be attributed to spatial variations of optical water properties affecting the attenuation coefficient $\alpha$. The cruise vessel's location on day 13 near the Midriff Islands is known for high phytoplankton concentrations, which enhances the absorption of shortwave radiation (Álvarez Borrego, 2012). Indeed, we can accurately model the maximum of $\Delta\mathrm{SST}$ on day 13 by increasing $\alpha$, but this reduces the model performance across the time series

overall.

Yet, even with constant parameters obtained from Bayesian inference, our model captures the observed variability of $\Delta\mathrm{SST}$ despite the heterogeneity of the MOCE-5 dataset. For idealized atmospheric modeling, we argue that one is usually interested in average sea properties for conditions of interest, or a parameter sensitivity experiment. The parameter posterior distribution reported here can be used directly as a prior distribution for a re-calibration based on relevant additional data.

**5.3   Inaptitude of slab ocean**

In contrast to the DiuSST and ZB05 models, the Slab model fails to capture the observed diurnal warming dynamics, including the wind dependence of $\Delta\mathrm{SST}$. This is because the evolution of slab temperature depends on wind speed only via the latent and sensible surface heat flux, whereas wind strongly influences vertical heat transport via the diffusion term in DiuSST or via the stability function in ZB05. In other words, the wind dependence of upper-ocean turbulence controls the effective heat capacity

of the upper ocean, but the slab's heat capacity is fixed by the slab thickness $h$.

One might argue that the slab exhibits too little variability because the parameter $\xi_1$, the inverse timescale of temperature relaxation, is set too large. Indeed, decreasing $\xi_1$ allows for stronger diurnal warming but also leads to excessive nighttime cooling and a slow temperature decline in the afternoon, thus worsening the agreement with the observations overall (see Fig. E1). In fact, the shape of the diurnal warming curve produced by a slab with small corrector coefficients $\xi_1$ and $\xi_2$ resembles

the diurnal temperature evolution observed at around $1\,\mathrm{m}$ depth, e.g., by moored buoys (Börner, 2021). This highlights that slab oceans mimic bulk SST rather than skin SST. Also other models of the diurnal SST cycle are sometimes calibrated with respect to bulk SST. However, skin SST is the relevant quantity for atmospheric studies, since the atmosphere only senses the temperature of the sea skin.

**5.4   Interactive diurnal SST in atmospheric simulations**

As a more realistic alternative to a slab model, the DiuSST and ZB05 models can be coupled to high-resolution cloud-resolving atmospheric models, e.g. to study how air-sea interactions impact atmospheric convection in idealized setups.

The ZB05 model appeals with its rigorous derivation from Monin-Obukhov similarity theory, condensed into one integrated bulk equation for the warm layer plus the cool skin layer scheme. This makes it computationally fast, though calculating



the skin layer depth $\delta$ involves an implicit equation that requires iterative solving (see Eqs. (5) and (6) in Zeng and Beljaars
(2005)). The empirical shape parameter $\nu$ of the vertical temperature profile could serve as a tuning parameter in sensitivity
experiments. While ZB05 matches fairly well with the observations used in this study, its performance has varied in other
comparative studies (Bellenger and Duvel, 2009; Jia et al., 2023). Running DiuSST requires integrating a discretized PDE,
making it slightly slower than ZB05 in the Python implementations used here[1] but not significantly – especially relative to the
cost of a coupled atmospheric component. Compared to ZB05 or similar existing diurnal warm layer schemes, modelers may
find DiuSST easier to tune, interpret and adapt to their purposes due to the conceptually simple modular structure. The fact
that DiuSST resolves the vertical temperature profile opens up opportunities to study, for example, how biological upper-ocean
processes interact with weather and climate across the air-sea interface.

Recently, we implemented our upper ocean model as an interactive sea surface boundary condition in the System for Atmo-
spheric Modeling (SAM) model (Khairoutdinov and Randall, 2003). At each horizontal grid point of the atmospheric model,
the surface boundary condition is independently updated at each time step by numerically integrating eq. (A1) to the next time
step, based on the local atmospheric forcing $\mathcal{F}$. This produces a responsive, spatially heterogeneous sea surface whose temper-
ature feeds back into the atmospheric boundary layer. According to first tests, the difference in computation time between this
setup and a coupled slab ocean is unnoticeable.

The influence of our model on the spatio-temporal patterns of convection is subject of future research. As a global kilometer-
scale simulation study by Shevchenko et al. (2023) suggests, diurnal warm layers can have a significant impact on the atmo-
sphere locally where diurnal warming is strong; yet their role for the large-scale tropical cloud field and convective organization
remains unclear.

## 6  Conclusions

This paper presents a simple, one-dimensional prognostic model of diurnal sea surface temperature variability in the tropical
ocean, described by Eq. (1) and formulated as a numerical scheme in Eq. (A1). Written as a single partial differential equation,
the model describes upper ocean heat transfer through three idealized terms, controlled by three tuning parameters: an eddy
diffusivity $\kappa_0$, a bulk mixing rate $\mu$ and an attenuation coefficient $\alpha$. $\kappa_0$ controls the strength of wind-driven turbulent heat
transport, $\mu$ determines the relaxation rate towards the foundation temperature and $\alpha$ specifies how deeply solar radiation
penetrates into the ocean.

First, we used Bayesian inference to estimate the values of these parameters based on an observational data set recorded on
the MOCE-5 cruise in the Eastern Pacific. Then, we compared the performance of our model with two other models of diurnal
SST dynamics: a slab ocean model, as previously used to mimic a responsive sea surface in atmospheric simulations, and the
diurnal warm layer scheme by Zeng and Beljaars (2005) (ZB05). Our results showed that slab models with fixed heat capacity
cannot capture diurnal SST variability realistically. Instead, our model reproduced an exponential dependence of the diurnal
warming amplitude on wind speed, in accordance with observations and similar to ZB05.

---

[1] Available at https://github.com/reykboerner/diusst.





By introducing a diffusion term that scales with surface wind stress, we proposed a simple solution that offers significantly improved results compared to a slab model and parameterizes the basic features of upper ocean turbulence in a physically interpretable way. Enhancing the model in the future, e.g. by refining the diffusivity profile to include effects of precipitation, is facilitated by the short modular code. To confirm the model's validity for diverse conditions, the Bayesian approach offers a natural way to update the parameter estimates based on additional data.

Due to its numerical and conceptual simplicity, we envision the model presented in this study to serve as a generic interactive boundary condition for oceans in idealized cloud-resolving simulations of the atmosphere. This will enable a wind-responsive SST field that evolves under and feeds back into the atmospheric fluid dynamics. Given the multiscale interaction between the diurnal cycle and large-scale patterns of tropical convection, it becomes increasingly clear that we must consider diurnal warm layer dynamics to better understand the mechanisms of marine cloud organization. Our work thus hopes to contribute to bridging the gap between idealized studies of convective aggregation and real-world process understanding.

*Code and data availability.* A documented Python implementation of the model code is available on https://github.com/reykboerner/diusst (Börner, 2024), along with additional code, an example to run the code, and the observational data. A DOI will be added to the repository upon acceptance of the manuscript.

*Video supplement.* A video presentation introducing the model is available at https://www.youtube.com/watch?v=KdOWF_fzRLE.

## Appendix A: DiuSST model – Discretized model equation

The discretized form of eq. (1), using an explicit Euler scheme in time and finite differences on a non-uniform grid in space, reads

$$
\frac{T_n^{i+1} - T_n^i}{\Delta t_i} = \kappa(z_n, t_i) \left[ \left( T_{n+1}^i - 2T_n^i + T_{n-1}^i \right) \cdot \left( \left. \frac{\mathrm{d}n}{\mathrm{d}z} \right|_{z_n} \right)^2 + \frac{T_{n+1}^i - T_{n-1}^i}{2} \cdot \left. \frac{\mathrm{d}^2 n}{\mathrm{d}z^2} \right|_{z_n} \right]
$$

$$
+ \left. \frac{\partial \kappa(z, t_i)}{\partial z} \right|_{z_n} \left[ \frac{T_{n+1}^i - T_{n-1}^i}{2} \cdot \left. \frac{\mathrm{d}n}{\mathrm{d}z} \right|_{z_n} \right] - \mu \frac{T_n^i - T_f}{|z_n - z_f|}
$$

$$
+ \frac{1}{\rho_w c_p} \left[ \left( Q_{n+1}^i - Q_n^i \right) \cdot \left. \frac{\mathrm{d}n}{\mathrm{d}z} \right|_{z_n} \right], \qquad (\text{for } n = 0, \ldots, N - 1), \tag{A1}
$$

where $n$ is the depth index and $i$ the time index (nomenclature as in the main text, see also table 1). Here $\kappa(z, t)$ is given by eq. (2); its derivative by $z$ is computed analytically. The derivatives $\mathrm{d}n/\mathrm{d}z$ and $\mathrm{d}^2 n/\mathrm{d}z^2$,

$$
\frac{\mathrm{d}n}{\mathrm{d}z} = \left[ \ln \epsilon \cdot \left( \frac{\Delta z_0}{1 - \epsilon} + z \right) \right]^{-1}; \qquad \frac{\mathrm{d}^2 n}{\mathrm{d}z^2} = \left[ -\ln \epsilon \cdot \left( \frac{\Delta z_0}{1 - \epsilon} + z \right)^2 \right]^{-1}, \tag{A2}
$$





map the finite differences onto the non-uniform grid spacing, where the stretch factor $\epsilon > 1$ solves the equation $\epsilon(\Delta z_0, N) = 1 + (1 - \epsilon^N) \Delta z_0 / z_f$ (see also eq. (9)). At the foundation depth, the Dirichlet boundary condition is $T_N^i = T_f$ for all $i$. At the surface ($n = 0$), we require a choice on dealing with the free boundary. We introduce a dummy grid point ($n = -1$) at $z_{-1} = \Delta z_0$, at which we set the temperature to $T_{-1}^i = T_0^i$ for all $i$ (essentially assuming that the temperature gradient vanishes

at the air-sea interface). The surface heat flux (at $n = 0$) is given by

$$Q_0^i := R_{\mathrm{sw}}(t_i) + R_{\mathrm{lw}}(t_i) + Q_{\mathrm{s}}(t_i) + Q_{\mathrm{l}}(t_i) \,. \tag{A3}$$

For $n > 0$, the heat flux $Q_n^i \equiv Q(z_n, t_i)$ is given by eq. (7). Using a forward difference for $Q_n^i$ ensures that the integrated heat flux over the domain corresponds to the total heat uptake within the diurnal layer. Note that the diffusion part of eq. (1) involves two terms in eq. (A1) owing to the chain rule of differentiation.

## Appendix B: Surface reflection of solar radiation in DiuSST

Based on Fresnel's equations, assuming unpolarized light, the reflected fraction $\mathcal{R}$ of irradiance incident on the (flat and smooth) air-sea interface is given by $\mathcal{R} = (\mathcal{R}_\perp + \mathcal{R}_\parallel)/2$, where the contributions from the two polarization directions read

$$\mathcal{R}_\perp = \left( \frac{n_a \cos\phi - n_w \sqrt{1 - (\frac{n_a}{n_w} \sin\phi)^2}}{n_a \cos\phi + n_w \sqrt{1 - (\frac{n_a}{n_w} \sin\phi)^2}} \right)^2 \tag{B1}$$

$$\mathcal{R}_\parallel = \left( \frac{n_a \sqrt{1 - (\frac{n_a}{n_w} \sin\phi)^2} - n_w \cos\phi}{n_a \sqrt{1 - (\frac{n_a}{n_w} \sin\phi)^2} + n_w \cos\phi} \right)^2 \,. \tag{B2}$$

Here $\phi$ is the solar angle with respect to the surface normal; $n_a$ and $n_w$ denote the refractive index of air and water, respectively (see table 1). Thus, we approximate the transmitted solar irradiance $R_{\mathrm{sw}}$ entering the water body as

$$R_{\mathrm{sw}}(t) = \left(1 - \mathcal{R}(\phi(t)\right) R_{\mathrm{sw},\downarrow}(t), \tag{B3}$$

where $R_{\mathrm{sw},\downarrow}$ is the downward shortwave irradiance above the sea surface.

## Appendix C: Alternative diffusivity profiles

In the main text, we introduce a highly idealized linear diffusivity profile $\varphi(z)$ (eq. (3), called LIN hereafter). Here we briefly discuss two alternatives that have been investigated in the course of this study. Consider the profile

$$\varphi(z,t) = \frac{1 - S(t)\sigma \exp(z/\lambda)}{1 - S(t)\sigma \exp(z_f/\lambda)} \,, \tag{C1}$$

given in terms of the suppressivity $\sigma \in [0,1]$, the trapping depth $\lambda > 0$, and the (time-dependent) stability function $S$.




First, set $S(t) = 1$ for all $t$, giving a time-independent exponential profile (EXP). If $\sigma > 0$, then $\varphi$ decreases exponentially
towards the sea surface, where $\varphi(0) = 1 - \sigma$. The parameter $\lambda$, setting the curvature of the profile, determines the depth scale
at which the diffusivity is suppressed. In the limit $\lambda \to \infty$, we obtain the linear profile in eq. (3).

Second, we investigate a state-dependent stability function (STAB) that depends on the vertical temperature gradient $\Delta T_\lambda(t) :=$
$T(0,t) - T(\lambda,t)$, given by

$$S(t) = \begin{cases} \min\left(1, \Delta T_\lambda(t)/T_{\text{strat}}\right) & \text{if } \Delta T_\lambda(t) > 0 \\ 0 & \text{otherwise,} \end{cases} \tag{C2}$$

where $T_{\text{strat}}$ is a reference temperature at which thermal stratification dominates over turbulent mixing. The physical rationale
behind this is that density stratification due to near-surface heat trapping locally enhances water column stability, inhibiting
vertical diffusion. As soon as the sea skin temperature cools again with respect to the temperature at depth $\lambda$, diffusivity
increases, mimicking enhanced afternoon mixing due to an unstable temperature profile.

Comparing simulations with the LIN and EXP profiles, we find that EXP performs slightly better than LIN for the ob-
servational data set considered in this study, particularly with regards to night-time skin cooling (Fig. C1). Though arguably
more realistic than LIN and EXP, the STAB profile does not necessarily perform better because it kills the coarse-grained skin
cooling effect in our model. Comparing the effect of EXP and STAB on the vertical temperature profile (Fig. C2), we see that
STAB counteracts near-surface temperature inversions, creating a mixed warm layer of near-constant temperature that better
matches observations Soloviev and Lukas (1997). However, STAB would have to be combined with a skin layer scheme as in
ZB05 in order to capture the sea skin temperature in the presence of skin cooling. Due to the overall similarity of modeled SST
warming between the LIN, EXP, and STAB versions, we choose to present the simplest option, LIN, in the main text.

## Appendix D: Bayesian inference

Consider a model $\mathcal{M}(\Theta)$ controlled by the parameter set $\Theta$. Given the data $\mathcal{D}$, the probability that a certain value of $\Theta$
represents the truth follows from Bayes' rule,

$$P(\Theta|\mathcal{D}) = \frac{P(\mathcal{D}|\Theta)P(\Theta)}{P(\mathcal{D})} . \tag{D1}$$

Here $P(A|B)$ denotes the conditional probability of $A$ given $B$. On the right-hand side, $P(\mathcal{D}|\Theta)$ is the *likelihood* of observing
the data under the assumption that $\Theta$ represents the true model. The *prior* probability $P(\Theta)$ quantifies our knowledge of the
parameter set before observing the data. Lastly, the denominator states the probability of the data being true, which is indepen-
dent of the choice of model and merely adds a normalization factor. Hence the *posterior* probability $P(\Theta|\mathcal{D})$ is proportional
to the product of the likelihood times the prior probability.

In practice, inferring the posterior distribution $P(\Theta|\mathcal{D})$ of the parameters $\Theta$ from the data $\mathcal{D}$ involves three steps: *1)* construct
the prior distribution $P(\Theta)$ in parameter space based on previous knowledge or belief; *2)* define a suitable likelihood function
$\mathcal{L}(\Theta) \equiv P(\mathcal{D}|\Theta)$; and *3)* compute the (un-normalized) posterior distribution by evaluating the product $\mathcal{L}(\Theta)P(\Theta)$.



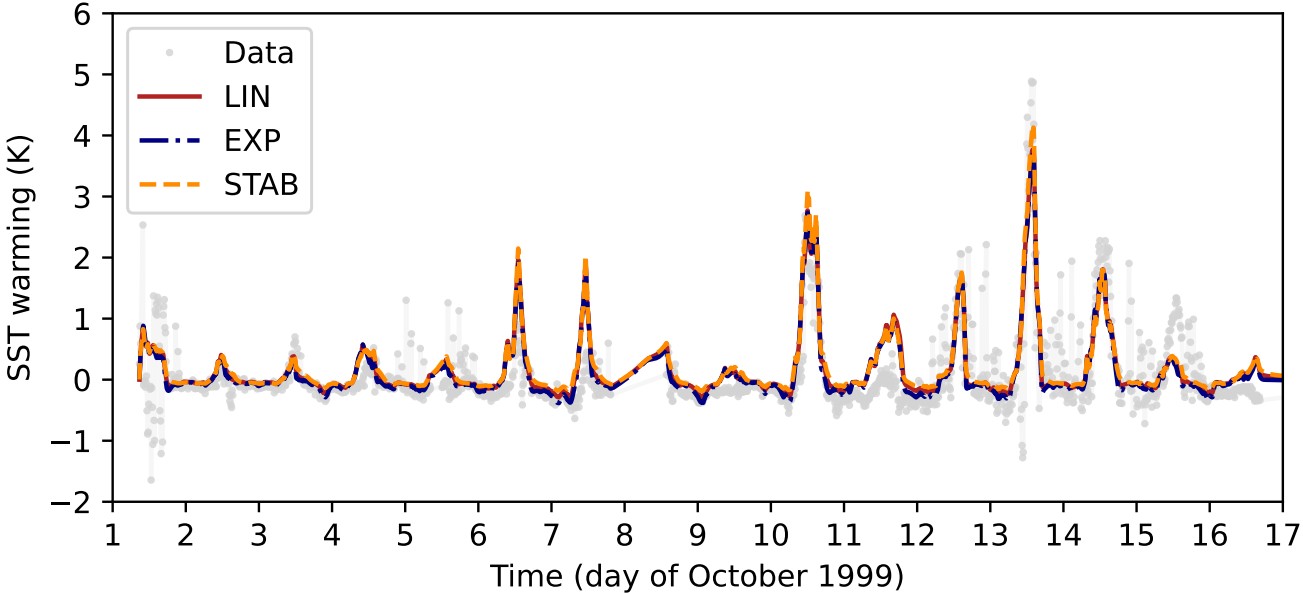

**Figure C1.** Comparison of modeled diurnal warming $\Delta$SST between different diffusivity profiles $\varphi$ discussed in C and the main text. LIN (solid red line) corresponds to the linear profile (eq. (3)) used throughout the main text. EXP (dash-dotted blue) and STAB (dashed orange) show results for the corresponding versions of $\varphi$ described in C. For each case, the model parameters $\kappa_0$, $\mu$, and $\alpha$ are calibrated specifically via Bayesian inference. All other model settings are kept constant, as given in table 1.

### D1 Choice of prior

Our sensitivity study under idealized forcing (section 3.1) provides orientation on physical ranges of the parameters $\{\kappa_0, \mu, \alpha\}$ of the DiuSST model. In the literature, common values of oceanic turbulent vertical diffusivity are on the order of $\kappa \sim$ 560 $10^{-4}\,\mathrm{m^2\,s^{-1}}$, varying with location and depth Denman and Gargett (1983). Stating a physical value for the attenuation coefficient $\alpha$ is difficult since, in reality, attenuation of shortwave radiation depends strongly on wavelength and the biochemical composition of the seawater. Empirical values for the diffusive attenuation coefficient of photosynthetically active radiation range from $\sim 10^{-2}\,\mathrm{m^{-1}}$ to $10\,\mathrm{m^{-1}}$ Son and Wang (2015). However, we note that the model parameters are conceptual, particularly the mixing coefficient $\mu$, such that they do not necessarily represent directly observable physical quantities.

Reflecting our limited knowledge, we impose uniform prior distributions for the parameters $\kappa_0$ and $\alpha$ but constrain their range to $\kappa \in [0, 5 \times 10^{-4}]\,\mathrm{m^2\,s^{-1}}$ and $\alpha \in [0.05, 10]\,\mathrm{m^{-1}}$. The parameter $\mu$ requires more subtle treatment because it acts mainly near the foundation depth, whereas the likelihood function evaluates temperature differences near the surface (see below). Fig. 2 indicates that excess heat remains in the interior of the diurnal layer if $\mu$ falls below $\sim 1 \times 10^{-4}\,\mathrm{m\,s^{-1}}$. Based on this insight, we define a normally distributed prior for $\mu$ with mean $6 \times 10^{-3}\,\mathrm{m\,s^{-1}}$ and standard deviation $1.5 \times 10^{-3}\,\mathrm{m\,s^{-1}}$.



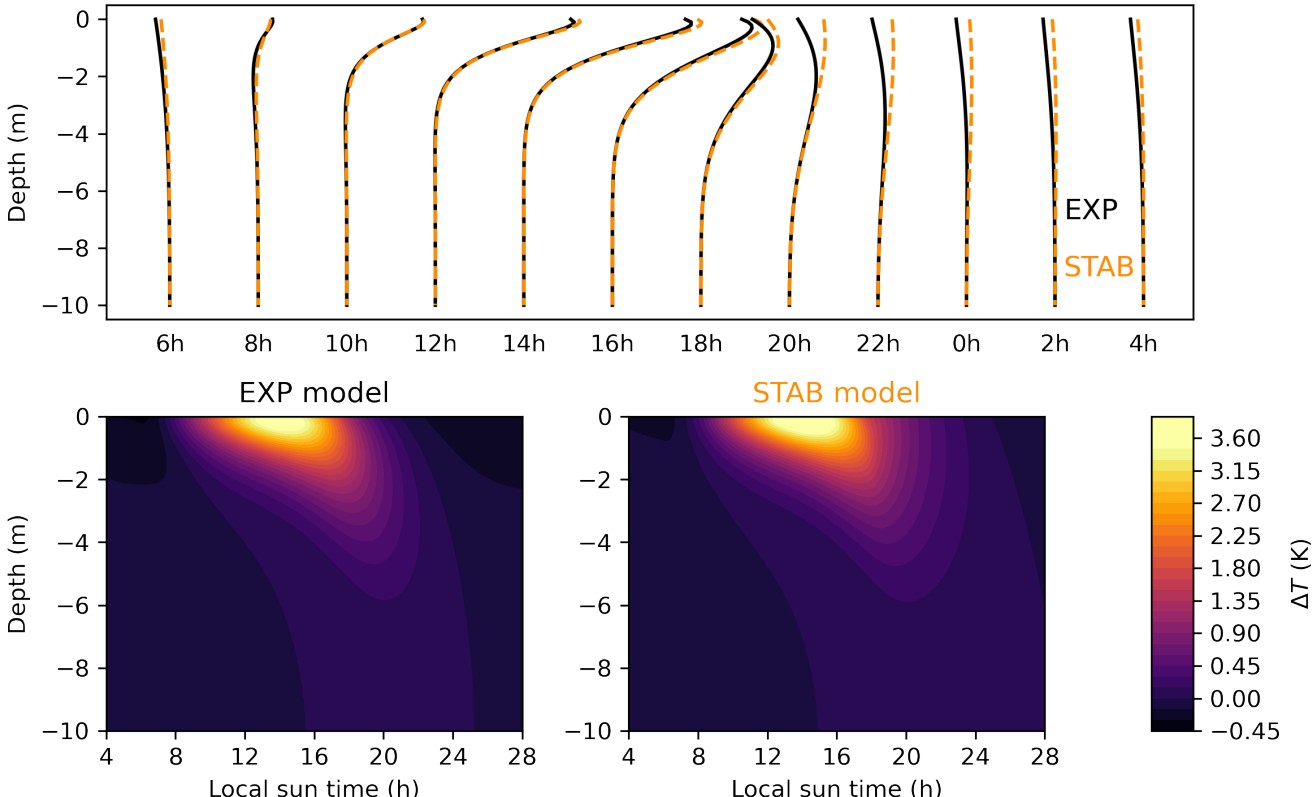

**Figure C2.** Comparison of vertical temperature profiles, produced by the model, with diffusivity profile EXP (dark blue) and STAB (orange). The top panel shows the vertical profile every two hours, shifted by $1\,\mathrm{K}$ per hour along the horizontal axis. The bottom panels show the temperature difference $\Delta\mathrm{T}$ with respect to the foundation temperature as a function of depth and time for the EXP model (left) and STAB model (right).

Thus the three-dimensional prior distribution $P(\Theta)$ is uniform and bounded along the $\kappa_0$ and $\alpha$ directions but Gaussian with respect to $\mu$.

Note that the posterior distribution with respect to $\mu$ differs clearly from the prior distribution, confirming that the data added information on the parameter $\mu$ (Fig. 4).

**D2   Defining the likelihood function**

Our *training data* $\mathcal{D}$ consists of a six-day subset of the diurnal warming time series from the MOCE-5 cruise, as indicated by the gray shading in Fig. 3. Specifically, we select days 2-4 and 13-15 of October 1999 as training data, while all other days from 1 to 16 October make up the *validation data*. By this choice, the training data contains warming events of different amplitudes, covering the observed range from a few tenths of a Kelvin up to $5\,\mathrm{K}$. Moreover, the two sub-intervals composing $\mathcal{D}$ correspond





to different geographical regions (open Pacific vs. Gulf of California), presumably featuring differing environmental conditions.
This ensures that the resulting parameter estimates will be valid for a broader range of conditions.

We choose a likelihood function that decays exponentially with the weighted square error between model and data,

$$\mathcal{L} \propto \exp\left(-\sum_j \frac{\left(\Delta\mathrm{SST}_\mathcal{M}(t_j) - \Delta\mathrm{SST}_\mathcal{D}(t_j)\right)^2}{\Sigma_j^2}\right), \tag{D2}$$

where the index $j$ runs through all data points in $\mathcal{D}$ and $\Delta\mathrm{SST}_\mathcal{M}$ ($\Delta\mathrm{SST}_\mathcal{D}$) denotes the modeled (observed) diurnal warming at the corresponding time of observation. Specifically, $\Delta\mathrm{SST}_\mathcal{M}(t_j) = T(0, t_j) - T(z_d, t_j)$, with $z_d$ being the depth of the vertical grid point closest to the observational reference depth $d = -3\,\mathrm{m}$. Furthermore, the standard error $\Sigma_j$ determining the relative
weight of the $j$-th data point is defined as

$$\Sigma_j = 2\epsilon_j(1 + v_j/v_{\max}), \tag{D3}$$

where $\epsilon_j$ denotes the uncertainty of the $3\,\mathrm{m}$ temperature measurement, as stated in the data set; $v_j$ is the current boat speed and $v_{\max}$ is the maximum boat speed throughout the data set. This choice reflects a reduced confidence in measurements taken while the ship was moving rapidly. Indeed, the MOCE-5 time series features several short, isolated warming spikes of around 1-2°C during nighttime hours, which cannot be explained by atmospheric forcing but could have been caused by the boat's movement.
Since the ship mainly moved at night while often remaining at fixed locations during the day, $\Sigma_j$ emphasizes daylight data.

### D3    Computing the posterior distribution

To approximate the posterior distribution, we perform Markov chain Monte Carlo (MCMC) sampling using the `emcee` package in Python Foreman-Mackey et al. (2013). It is based on the affine-invariant GW10 algorithm proposed by Goodman and Weare 2010. The algorithm explores the parameter space with multiple interdependent walkers whose moves efficiently deal with
correlations, yielding fast convergence.

At each step of the Markov chain, the likelihood function, Eq. (D2), is evaluated for each walker at its respective position $\Theta$ in parameter space. Each evaluation requires simulating the model for the duration of the training data, which makes the sampling computationally expensive. For each of 24 walkers we generate 4000 steps, initialized at the value $\{\kappa_0, \mu, \alpha\} = \{10^{-4}\,\mathrm{m}^2\,\mathrm{s}^{-1}, 6\times10^{-3}\,\mathrm{m}\,\mathrm{s}^{-1}, 4\,\mathrm{m}^{-1}\}$. Finding an auto-correlation length of around 50 steps, this gives around 80 independent
samples.

We summarize the resulting parameter estimates in Tab. D1, listing common statistical estimators as obtained from the posterior distribution of the DiuSST (Fig. 4) and Slab (Fig. D1) models, respectively. Bayesian inference for the Slab model was performed in analogy to the DiuSST model but in a four-dimensional parameter space $\{h, S, \xi_1, \xi_2\}$ and with the slab temperature anomaly representing $\Delta\mathrm{SST}_\mathcal{M}$ in the likelihood function. The run was initialized at values $\{h, S, \xi_1, \xi_2\} = \{1, 100, 10^{-4}, 10^{-9}\}$
with uniform priors for $h$, $\xi_1$, $\xi_2$ and a Gaussian prior for $S$ with mean $100\,\mathrm{W}\,\mathrm{m}^{-2}$ and standard deviation $10\,\mathrm{W}\,\mathrm{m}^{-2}$.





| Model | Parameter | MAP | Mean | Median |
|---|---|---|---|---|
| DiuSST | $\kappa_0$ $(\mathrm{m^2\,s^{-1}})$ | $1.34 \times 10^{-4}$ | $1.41 \times 10^{-4}$ | $1.40 \times 10^{-4}$ |
| | $\mu$ $(\mathrm{m\,s^{-1}})$ | $2.85 \times 10^{-3}$ | $3.00 \times 10^{-3}$ | $2.96 \times 10^{-3}$ |
| | $\alpha$ $(\mathrm{m^{-1}})$ | 3.52 | 3.83 | 3.70 |
| Slab | $h$ (m) | 1.20 | 1.20 | 1.20 |
| | $S$ $(\mathrm{W\,m^{-2}})$ | 92.67 | 93.86 | 93.94 |
| | $\xi_1$ | $1.19 \times 10^{-4}$ | $1.20 \times 10^{-4}$ | $1.18 \times 10^{-4}$ |
| | $\xi_2$ | $3.1 \times 10^{-11}$ | $5.3 \times 10^{-11}$ | $3.7 \times 10^{-11}$ |

**Table D1.** Parameter estimates obtained from the sampled posterior distribution.

## Appendix E: Slab model parameter sensitivity

As mentioned in the main text, the Bayesian optimization procedure applied to the Slab model acts mainly by ramping up the relaxation parameter $\xi_1$, which constrains diurnal warming to $1\,\mathrm{K}$. Larger diurnal amplitudes are achieved by reducing $\xi_1$ (see Fig. E1). However, in that case the slab also cools excessively at night, and the diurnal warming peak shifts into the late afternoon, with a slow temperature decrease after peak warming.

## Appendix F: Model comparison over full MOCE5 time series

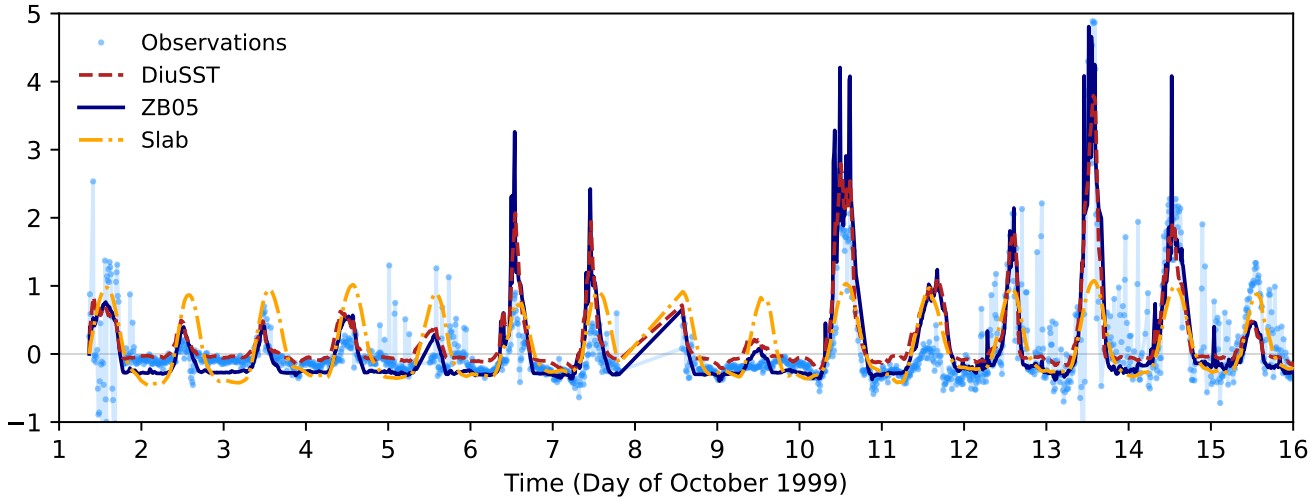

**Figure F1.** Comparison of observed diurnal warming $\Delta$SST (light blue) against the predictions of the calibrated DiuSST (red dashed), ZB05 (dark blue), and calibrated Slab (orange dash-dotted) models, showing the full 16-day time series recorded on the MOCE-5 cruise.





**Figure D1.** Posterior distribution for the parameters of the slab model, obtained from Bayesian MCMC sampling. Compare with Fig. 4, which shows the results for the DiuSST model.





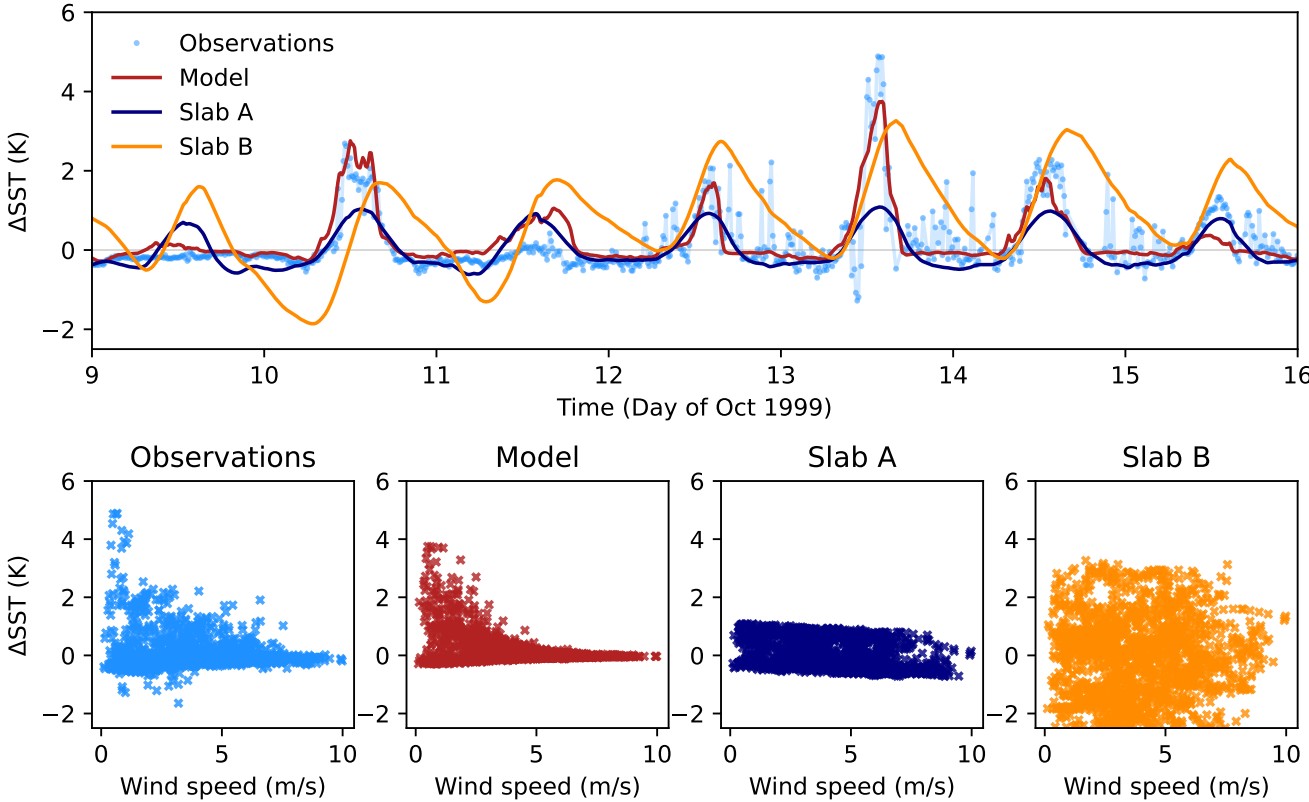

**Figure E1.** Performance of the slab model as a function of its control parameter $\xi_1$. The solid red (Model) and blue (Slab A) lines show simulations of our calibrated model and the slab model, respectively, with parameter values given in table D1. The orange curve (Slab B) shows a slab simulation with $h = 1.1\,\mathrm{m}$, $S = 70\,\mathrm{W\,m^{-2}}$, $\xi_1 = 1 \times 10^{-7}$ and $\xi_2 = 0$. Note that for Slab B, $\xi_1$ is three orders of magnitude smaller compared to Slab A.

*Author contributions.* JOH and RF conceived and supervised the study. All authors developed the model and methodology. RB produced the code, acquired and curated the data, conducted the investigation and formal analysis, performed the model evaluation and validation, and prepared the figures. RB prepared the manuscript with input from all co-authors.

*Competing interests.* The authors declare no competing interests.

*Acknowledgements.* We thank Gorm Gruner Jensen, Peter Ditlevsen, and Chong Jia for useful discussions, as well as Peter Minnett for providing the MOCE-5 dataset and helpful feedback on the manuscript. We further thank the two anonymous reviewers for their constructive feedback which helped strengthen the manuscript. The development and deployment of the instruments used during the MOCE-5 cruise was



funded by NASA. R. F. and J. O. H. gratefully acknowledge funding from the Villum Foundation (grant no. 13168). J. O. H. acknowledges
funding from the European Research Council under the European Union's Horizon 2020 Research and Innovation programme (grant no.
771859) and the Novo Nordisk Foundation Interdisciplinary Synergy Program (grant no. NNF19OC0057374). R.B. acknowledges funding
from the European Union's Horizon 2020 research and innovation programme under the Marie Sklodowska-Curie grant agreement no.
956170 (CriticalEarth).



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
