# Peer review of "DiuSST: A conceptual model of diurnal warm layers for idealized atmospheric simulations with interactive SST"

_EGUsphere, 2024_

## Author Response (AR1)

**Revised submission: egusphere-2024-1876**

Dear Editors,

Thank you for considering our manuscript for publication in GMD. We are grateful to the two reviewers for their time and useful feedback. In the following, please find a detailed response (in blue text) to the reviewers' comments (black text), with changes implemented in the revised manuscript noted in green.

Kind regards,

Reyk Börner
on behalf of all coathors

**Reviewer #1**

General comment:

This paper presents a new modeling framework to represent the near-surface ocean and its temperature diurnal cycle. By enabling the representation of the temperature profile in the first meters of the ocean, this model shows promise for future applications to study physical interactions between the ocean surface and the atmosphere, between the ocean surface and the ocean interior and in particular processes involving biogeochemical cycles. As exposed in the article, there is a growing interest in understanding and modelling ocean-atmosphere coupled mechanisms at the sub-diurnal time-scale among which these involving atmospheric deep convection (triggering, aggregation). This study is therefore timely and has a high scientific significance. The scientific quality of the approach is excellent with hypotheses being in general discussed clearly and confronted to in situ observations when possible. The article proposes a detailed and complete view of the modeling framework and remains synthetic and clear in the comments of the results and in the discussions.

In conclusion, this is an overall excellent and convincing article which I will be glad to recommend for publication in Geoscientific Model Development after some specific comments have been addressed.

We thank the reviewer for their evaluation and are glad that they recommend our article for publication. We will respond to the specific comments after each point below.

Specific comments:

1- One of the advantages of the proposed model is to represent the evolution of the temperature profile within the first meter of the ocean. However, no attempt to validate these profiles against observations is made. Yet, in situ observations exist for the same cruise

(Ward et al 2006 cited in the article). If these data are not available to the authors, the POSH parameterization of Gentemann et al. (2009, cited in the draft) could be of some help to validate DiuSST profiles. This validation could also help the reader see the advantage of the present model compared with the Zeng and Beljaars parameterization that assumes a very steep profile of temperature close to the interface.

We agree that a validation of the vertical profiles produced by DiuSST would strengthen the article and further highlight the advantage of a depth-resolving model like DiuSST over Zeng & Beljaars 2005 (beyond the fact that in DiuSST the profile emerges as the solution of the wind-dependent diffusion equation whereas ZB05 assumes a fixed functional form of the vertical profile).

We have attempted to obtain the SkinDeEP data (Ward, 2006) which contains vertical profiles measured during the MOCE-5 cruise (the field campaign we used for calibrating and validating our model). Unfortunately, after corresponding with the author of the dataset, we must conclude that the data are currently not available for this review process, though the author is open to sharing the data for a potential collaboration on a standalone publication.

As suggested by the reviewer, we have compared vertical profiles of DiuSST with the normalized profiles of the POSH model (Gentemann et al. 2009), shown below (Fig. II) for the days October 3, 10, and 13 of the MOCE-5 field study. We have set the surface and foundation temperature of the POSH profiles to that of the DiuSST simulations, since we are interested in the shape of the vertical profile, not the magnitude. Besides wind speed, which we took from the observations, the key variable in the POSH parameterization is the warm layer depth D_L. We compute D_L as described in Gentemann et al. (2009) and Fairall et al. (1996), setting the thermal expansion coefficient to and computing the wind stress $\tau$ from wind speed $u$ as

$$\tau = \rho_a C_D u^2$$

where $\rho_a$ is the air density and $C_D = 1.3 \cdot 10^{-3}$ is the drag coefficient.

Gentemann et al. (2009) added several corrections to their model to reduce the error of modelled surface warming compared to observations. These corrections include an enhancement of the absorption of shortwave radiation and a reduction of integrated heat and momentum fluxes. The corrections could be specific to the observations considered and may not apply generally. For the MOCE-5 observations, we found that the warm layer depth D_L during midday is very shallow with these corrections for various wind speeds (see Fig. I below). If we set the correction parameter $\varepsilon_\tau = -2.5 \cdot 10^{-5}$ in Eq. (16) of Gentemann et al. (2009), whereby we enhance wind shear stress, the evolution of D_L is closer to our expectation (though perhaps a bit deep at noon for low wind speeds, e.g. on Oct 13). With this change, the DiuSST and POSH profiles are comparable (see Fig. II below).

[Figure]

Fig. I. Warm layer depth over time for selected days based on the formula (6) in Gentemann et al. (2009) with heat fluxes and wind stress taken from the MOCE-5 observations.

[Figure]

Fig. II. Comparison between DiuSST (black) and POSH (orange) profiles for the three selected days.

Since D_L, and thus the profiles of the POSH model, are quite sensitive to the corrections described above, we conclude that observations are needed to adequately validate/calibrate DiuSST profiles. Therefore, we would opt against adding the analysis described here to the revised manuscript, unless the reviewers or editors disagree.

Added paragraph in section 5.2 of the manuscript:

"To validate the vertical profiles generated by DiuSST, depth-resolved observations are needed. Incorporating information at depth in the calibration can further constrain the parameter estimates, help resolve apparent parameter correlations, and inform the choice of diffusivity profile $\varphi$. However, the vertical temperature profiles measured during the MOCE-5 cruise \citep{ward_near-surface_2006} are subject to various other influences besides diurnal warming \citep{gentemann_profiles_2009}, and currently not publicly available. Validating the profiles is thus left for future work."

2- The authors could precise a bit more the Markov Chain Monte Carlo approach. They argue that they can produce 80 independent samples out of a timeseries containing only 13 full diurnal cycles (Fig. F1). Correlations exist between the different parameters due to the correlation in the training data between the wind speed and shortwave radiation (line 281-284). One is left to wonder if the observation sample is sufficient for a robust estimation of the parameters and for the model validation.

We thank the reviewer for raising this point, which can be better clarified. The sampling refers to the random walks in parameter space to approximate the posterior distribution, following the emcee algorithm based on Goodman and Weare (2010). This algorithm evolves an ensemble of dependent random walkers in a way that deals particularly well with parameter correlations. The value of 80 independent samples comes from dividing the total length of the resulting Markov chain (excluding a transient burn-in period) by its autocorrelation length, yielding an estimate for the number of independent samples of the posterior distribution in parameter space. This does not refer to the autocorrelation of the diurnal warming timeseries.

In fact, as mentioned in the manuscript, the posterior distribution is trained on 6 days of the diurnal warming timeseries, keeping the remaining days for validation. Of course, more training data would in principle be better but here we believe this is the best trade-off given the limited size of the observational record. As we point out in the text, the Bayesian inference framework we used allows iteratively updating the parameter estimates as more data becomes available (by using the previous posterior as a new prior).

We added "(along the Markov chain)" in section D3 to clarify that the autocorrelation length refers to the Markov chain in parameter space and not the observational timeseries.

3- With a vertical resolution of 0.1m, the diffusive microlayer of less than a millimeter thickness that is responsible for the cool skin phenomenon is not resolved by DiuSST. This is somehow acknowledged by the authors who speak about a "coarse-grained" cool skin (see discussions lines 295-297, 310-312, 412-416, 540-545). Indeed, static instabilities with vertical extend of the order of 0.1m are larger than the Kolmogorov scale and should lead to convective instability (Saunders 1967, Fairall et al. 1996, Zeng and Beljaars 2005). Therefore, one may argue that the STAB version of the model (instead of LIN, Appendix C)

is the most physical and should be used as the main model to represent the subskin temperature. In order to get a proper representation of the interfacial temperature, one would have to add a parameterization for the cool skin such as Fairall et al (1996).

*Saunders, P. M. (1967), The temperature at the ocean-air interface, J. Atmos. Sci., 24, 269 – 273*

We agree with the reviewer's argument that the STAB parameterization of the diffusivity profile would be more physical than LIN under the condition that a cool skin parameterization as in e.g. Fairall et al. 1996 be added. Without an additional cool skin scheme, however, the STAB version overestimates diurnal warming at night (see Fig. C1 of the original submission), which is why we opted for the LIN version in the main text following the principle of conceptual simplicity.

Technical comments:

Line 87-93: An issue of the Fairall et al. (1996) parameterization is the need to set to zero the diurnal warm layer during the night.

Thank you for noting this point.

Lines-160-165: When implemented in a model, downwelling longwave radiation can be an input rather than computed from the 10-meters height air temperature. This would help to take into account clouds effects for instance.

We will mention this in the revised version. Indeed, the formulae for computing surface fluxes (eq. 5) are only needed when these fluxes are not directly available as input data.

We added after eq. (5): "When coupled to an atmospheric model, the downward longwave radiative flux can be taken from the atmospheric model output."

Line 173: Can you illustrate the importance of taking into account the refraction angle?

Yes, we will add this to Appendix B (now Appendix C) in the revised version.

Added Fig. C1 of the revised manuscript, with a description:

"When neglecting surface reflection and the effect of surface refraction on the absorption of downward shortwave radiation, the simulated $\Delta$SST (under the same parameter settings) can be larger by up to \SI{0.2}{K} for the MOCE-5 case study (Fig. \ref{fig:noreflect}). However, note that if we would have calibrated the model without reflection and refraction, this difference would likely be compensated by adjusting the parameters."

Lines 190-195: By how much vary the integration timesteps? Can you provide some statistics, minimum and maximum?

We set the maximum time step to be 10s. The minimum time step is 0.004 seconds, though the number of time steps of less than a second makes up less than 0.15% of all steps. The mean time step is 6.4s, the median 5.7s.

Added sentence: "In the model simulations of this study, $\Delta t_i$ ranged from 0.004 to 10 s with a median time step of 5-6 s."

Line 195: About the limitation of the wind speed: Can you note the maximum diurnal amplitude of the SST at wind higher than 10 ms$^{-1}$?

At 10 m/s, the observed maximum diurnal amplitude is very close to zero, as seen in Fig. 7. The MOCE-5 dataset does not contain any data points with wind speed above 10 m/s.

Line 205: Is this 2K oscillation of the air temperature also visible in the MOCE-5 data? Why don't you set it to a constant?

In the MOCE-5 data, the air temperature exhibits a diurnal cycle with an amplitude on the order of 1K (when looking at hourly values averaged over all days of observation for a given hour of day), though the maximum is in the afternoon. We argue that a periodic diurnal cycle in air temperature is a reasonable idealization of clear, sunny days as prevalent during the MOCE-5 cruise. However, we agree that in this idealized sensitivity study we could have also set the air temperature to a constant.

Added a footnote: "This amplitude is about twice the diurnal amplitude recorded in the observational dataset of this study, where the air temperature oscillates by around \SI{1}{K} with the maximum in the afternoon."

Line 324 : Is $T$ really the difference relative to $T_f$? The difference between them actually appears in the sur second and third terms of (15).

Thank you for spotting this error; we will correct eq. (15) accordingly.

Corrected.

Line 344 : noa?

This was an error with a reference which has been corrected for the revised manuscript.

Corrected.

Figure 8 legend: Wind dependence of the « maximum » or « peak » or « amplitude of the » diurnal warming.

We agree with this suggestion and will change the legend to "Wind dependence of the amplitude of diurnal warming".

Changed.

Lines 415 : Tu and Tsuang (2005) used 100 mm resolution close to the surface to produce a cool skin with a unidimensional model.

*Tu C. Y. and B. J. Tsuang, 2005: Cool-skin simulation by a one-column ocean model, Geophys. Res. Lett.,. 32, L22602, doi:10.1029/2005GL024252, 2005*

Thank you for pointing out this reference. However, that study states that it used 100 μm (not mm) close to the surface, so it has a much higher near-surface resolution compared to our study.

Lines 445-452 : Can one conclude that OGCMs with 1m resolution close to the surface (e.g. Bernie et al. 2005) suffer from the same problem than the slab model, producing very regular diurnal cycles?

*Bernie, D. J., S. J. Woolnough, J. M. Slingo, and E. Guilyardi, 2005: Modeling diurnal and intraseasonal variability of the ocean mixed layer. J. Climate, 18, 1190–1202.*

Yes, our results suggest that this is the case (assuming the temperature dynamics of the top ocean layer is governed by a bulk heat budget equation).

Added a sentence: "This issue also pertains to ocean models with a surface layer of a meter thickness or more, which may reproduce a mean diurnal cycle but not the full range of diurnal variability \citep{voldoire_assessment_2022}."

Line 465 : The fact that DiuSST resolves the vertical profile makes it also suitable for an inclusion in an ocean model (or a coupled model) with a resolution of 1 meter or coarser close to the surface. For this kind of application, the warming of the first level of the ocean model should be added to the resolved foundation temperature to compute the fluxes such as Voldoire et al 2022 and Bellenger et al. 2023 (already cited in the text).

*Voldoire, A., R. Roehrig, H. Giordani, R. Waldman, Y. Zhang, S. Xie, and M.-N. Bouin, 2022, Assessment of the sea surface temperature diurnal cycle in CNRM-CM6-1 based on its 1D coupled configuration, Geosci. Model Dev., 15, 3347–3370, DOI:10.5194/gmd-15-3347-2022.*

Thank you for this comment and reference, which we may incorporate in the revised manuscript. We agree that, in the manner the reviewer suggests, DiuSST could be included in an ocean model to have a finer vertical resolution near the surface and thus better represent air-sea interactions.

Added citation in the context of the previous comment.

Annex F : Maybe add some comments?

This is a valid point; we can provide a short comment on Fig. F1 in the revised version.

Moved this section in the Appendix up (now Appendix B) and added commentary.

**Reviewer #2**

Review of "DiuSST: a conceptual model of diurnal warm layers for idealized atmospheric simulations with interactive SST" by Borner et al.

This paper is very well written and presents a novel and simple model of diurnal SST

warming. It is insightful, and as such, I think it should be published subject to one major comment and some minor modifications.

We thank the reviewer for taking the time and for their positive assessment. We will respond to the comments point-by-point below.

Major comment

I was intrigued that the training data included most of the extremes (minima on 2-4 October, maximum on 13 October). I was wondering if this made the exercise "too easy", in that you did not have to independently simulate the extremes. What would happen if you trained on 5th-12th October – would the extremes on other days be well simulated?

I understand that to create the best and most useful operational model, including extremes in training is necessary. My comment is mainly out of scientific curiosity – if you don't train on extremes, can you simulate extremes well?

Thank you for this question. The sensitivity to the choice of training interval has been explored to a limited extent in the thesis by Börner (2021), where Bayesian inference was performed when training only on data from the open Pacific Ocean (days 2, 3 and 4, see section 6.1 of that reference). When training only on days with small diurnal warming amplitudes, the large diurnal warming events (e.g. 13 October) are not well captured (see figure below). This is because the estimated diffusivity and attenuation coefficient are then both about five times smaller compared to the estimates in the manuscript.

Arguably, the training set used in the manuscript is already small, and omitting too much information will yield unsatisfactory results. In principle, the model could capture events outside its training range, but it is important to find the right balance of the model parameters $\kappa$, $\alpha$ and $\mu$. Since all three parameters influence the amplitude of diurnal warming, a certain amount of data is necessary to disentangle their effects (is the warming due to high insolation or low diffusivity?). To better constrain parameter estimates based on a shorter training time interval, it would also help to include data about the vertical temperature profile, particularly for estimating $\mu$.

If requested, we can re-train the model on the interval 5-12 October and share the results with a revised manuscript.

Börner, R. (2021). Modeling diurnal sea surface warming in the tropical ocean, https://nbi.ku.dk/english/theses/masters-theses/reyk-borner/boerner_MSc_thesis.pdf

[Figure]

Minor comments:

Line 3. "strongly, and delicately" seems like a contradiction!

Indeed, this seems like a contradiction, but it is an intended oxymoron here to highlight that the influence of diurnal SST variations on the atmospheric moisture field is not straight-forward, even if it can have a significant impact. For more on this, see e.g. Jensen et al. (2022).

Jensen et al. (2022), The Diurnal Path to Persistent Convective Self-Aggregation. https://doi.org/10.1029/2021MS002923

Line 43. Regarding imprint of SST on the moisture field, and, more generally, the atmosphere boundary layer, there is a large body of literature, e.g. reviewed in Seo et al. 2023.   See also Skyllingstad et al. 2019.

Thank you for citing these references, which we are happy to include where appropriate.

Added these references in the introduction.

Line 80-81 "generally not physics-informed" seems too strong. I looked at just one of the papers listed, and it was physics-informed (Price et al.)

This is a fair point. While some of the models proposed in Price et al. 1987 are purely empirical, others also incorporate physical insight. We propose to simply change "generally" to "sometimes".

Changed as proposed.

Lines 161-162. There are lots of other major references like Large and Yeager (2009), Fairall et al. 1996, 2003, Edson et al. 2013…

Thank you for pointing to these references. We will review our reference list and update it accordingly.

Added Large and Yeager (2009) in the suggested place and Edson et al. (2013) where we mention the wind stress dependence on the square of the wind speed, as that seemed most appropriate. The other two suggested references are already cited in the paper.

Equation 10: max(2…   )

Thank you for the suggestion. We will add parentheses in eq. (10) for clarity.

Added parentheses.

Line 193. What are typical timesteps of the model?

As mentioned in the response to review #1: We set the maximum time step to be 10s. The minimum time step is 0.004 seconds, though the number of time steps of less than a second makes up less than 0.15% of all steps. The mean time step is 6.4s, the median 5.7s.

Added a sentence on this in section 2.4:

"In the model simulations of this study, $\Delta t_i$ ranged from 0.004 to $\SI{10}{s}$ with a median time step of 5-$\SI{6}{s}$."

Line 241. Do you think there is any sensitivity to varying qv ?

Yes, there is. The figure below shows the simulated surface warming (ΔSST) when assuming a doubled (halved) specific humidity q_v. As expected, higher q_v leads to higher surface temperatures since the heat loss due to the latent heat flux is reduced.

To the editors: Should this figure be added to the Appendix?

[Figure]

Line 357 "on days 6 and 7 (not shown)"

Actually, this is shown in Fig. F1. We will add a cross-reference.

Added cross reference to what is now Fig. B1.

Line 373 "moisture and momentum"

Thank you for the suggestion, we will add "and momentum". We assume that the reviewer refers to the influence of SST on the wind field, which in turn affects the momentum transfer at the air-sea interface.

Added "and momentum".

Lines 423-439 discuss the applicability of the model to different environments, but only within the eastern Pacific region of Fig. 3. Can you say anything about its applicability to other Tropical and non-Tropical regions? I am not asking for any modification of the model, but maybe you can say whether the background conditions in other regions will make the current model suitable or non-suitable.

Thank you for raising this, as it requires further clarification. We focus on the environmental conditions in the case study region (Eastern Pacific, Gulf of California) since we have a direct comparison between DiuSST and observations there. In developing the model, however, we have not assumed any properties specific to that region, or the tropics in general. We thus believe that our model would also be suitable for other regions, and the three model parameters could be tuned to reflect the optical and dynamical properties there.

Of course, there are many details our simplified model neglects, such as precipitation effects, wave breaking and tidal currents, which might play an increased role in other regions. The presence of sea ice at high latitudes might be another influential factor that is excluded from our model.

Added two sentences to clarify this point:

"At the same time, the heterogeneity of the training data presents a benefit, as it reduces the risk of overfitting and promises a more generally applicable calibration."

"While the present calibration could be specific to the background conditions of the study region, the model formulation itself is not."

Refs

Seo et al. 2023: Ocean Mesoscale and Frontal-scale Ocean-Atmosphere Interactions and Influence on Large-scale Climate: A Review., J. Clim. 10.1175/JCLI-D-21-0982.1

Skyllingstad et al. 2023: DOI: https://doi.org/10.1175/JAS-D-18-0079.1 .

---

## Author Response (AR2)

5 January 2025

**egusphere-2024-1876: Author's response, second review**

Dear Editors,

Thank you for accepting our manuscript subject to technical corrections. Below we address the remaining comment of reviewer 2.

The comment was:

*Regarding your reply about training the data on other time periods (e.g. periods with no extremes), I think this information would be useful to the reader.*

*Could you put a 1-2 sentence summary of your response to the question in the manuscript, and reference the MSc thesis? (Along the lines of "Note it is important to train on a period including extremes - when this was not done the model performance was notably poorer (ref)"...*

We thank the reviewer for this suggestion and have added the following two sentences accordingly in section 5.2 of the revised manuscript, third paragraph:

*It was important to train on a period including both extremes, i.e. small and large diurnal warming events, since otherwise the model performance was significantly poorer (see \citet{borner_modeling_2021}, section 6.1). Since all three parameters influence the amplitude of diurnal warming, a certain range of data is necessary to disentangle their effects (for instance, is it warming due to high insolation or low diffusivity?).*

Furthermore, we added a missing axis label in Fig. B1. Otherwise, no further changes were made to the manuscript.

Your sincerely,
Reyk Börner

on behalf of all authors